# Unified PAC-Bayesian Study of Pessimism for Offline Policy Learning with Regularized Importance Sampling

Imad Aouali[1,2]        Victor-Emmanuel Brunel[1]        David Rohde[2]        Anna Korba[1]

[1]CREST, ENSAE, IP Paris, France
[2]Criteo AI Lab, Paris, France

## Abstract

Off-policy learning (OPL) often involves minimizing a risk estimator based on importance weighting to correct bias from the logging policy used to collect data. However, this method can produce an estimator with a high variance. A common solution is to regularize the importance weights and learn the policy by minimizing an estimator with penalties derived from generalization bounds specific to the estimator. This approach, known as pessimism, has gained recent attention but lacks a unified framework for analysis. To address this gap, we introduce a comprehensive PAC-Bayesian framework to examine pessimism with regularized importance weighting. We derive a tractable PAC-Bayesian generalization bound that universally applies to common importance weight regularizations, enabling their comparison within a single framework. Our empirical results challenge common understanding, demonstrating the effectiveness of standard IW regularization techniques.

## 1 INTRODUCTION

Offline contextual bandits [Dudík et al., 2011] have gained significant interest as an effective framework for optimizing decision-making using offline data. In this framework, an agent observes a context, takes an action based on a policy, i.e., a probability distribution over a set of actions, and receives a cost that depends on both the action and the context. These sequential interactions, recorded as logged data, serve two purposes in offline scenarios. The first is off-policy evaluation (OPE), which aims to estimate the expected cost (risk) of a fixed target policy using the logged data. The second is off-policy learning (OPL), whose goal is to find a policy that minimizes the risk. In general, OPL relies on OPE's risk estimator.

In OPE, a significant portion of research has focused on the inverse propensity scoring (IPS) estimator of the risk [Horvitz and Thompson, 1952, Dudík et al., 2011]. IPS employs importance weights (IWs), which are the ratios between the target policy and the logging policy used to collect data, to estimate the risk of the target policy. Although IPS is unbiased under mild assumptions, it can suffer from high variance, especially when the target and logging policies differ significantly [Swaminathan et al., 2017]. To address this issue, various methods have been developed to regularize IPS, primarily by transforming the IWs [Bottou et al., 2013, Swaminathan and Joachims, 2015a, Su et al., 2020, Metelli et al., 2021, Aouali et al., 2023a, Gabbianelli et al., 2024]. While these regularizations introduce some bias, they aim to reduce the estimator's variance. Most of these IW regularizations have been proposed and investigated in the context of OPE, where the primary goal is to enhance the estimator's accuracy, typically measured by mean squared error (MSE). In contrast, off-policy learning (OPL) aims to find a policy with minimal risk. Therefore, it is crucial to determine whether these IW regularizations lead to better performance in OPL.

A common approach in OPL is to learn the policy through pessimistic learning principles [Jin et al., 2021], where the estimated risk is optimized along with a penalty term often derived from generalization bounds. Consequently, previous studies on OPL with regularized IPS estimators have adopted this approach but focused on specific IW regularizations. For example, Swaminathan and Joachims [2015a] studied the IPS estimator with clipped IWs and proposed learning a policy by minimizing the estimated risk penalized with an empirical variance term. Similarly, London and Sandler [2019] suggested an alternative regularization for the same estimator, incorporating an $L_2$ distance to the logging policy. Additionally, Sakhi et al. [2023] derived tractable generalization bounds for a simplified doubly robust version of the IPS estimator with clipped IWs, using these bounds for their pessimistic learning principle. Similarly, Aouali et al. [2023a] derived a tractable bound for an estimator

that exponentially smooths the IWs instead of clipping them and proposed two learning principles: one where the bound is optimized and another heuristic inspired by it. Finally, Gabbianelli et al. [2024] introduced implicit exploration regularization, where a constant is added to the denominator of the IWs and used a learning principle that directly minimizes the corresponding estimator since their generalization upper bound did not depend on the target policy.

A limitation of these studies is that their guarantees and learning principles are specific to the particular IW regularization they consider and are not transferable to other IW regularizations. Consequently, in their OPL experiments, IW regularizations are compared using different learning principles, making it difficult to determine if better performance is due to the enhanced properties of the proposed IW regularizer or merely an artifact of the proposed learning principle. As a result, it remains unclear whether a particular IW regularization yields better performance in OPL. This highlights a gap in the literature: there is no unified study providing bounds on the risk of policies learned using pessimistic learning principles tailored to various regularized IW estimators of the risk. Our work aims to bridge this gap. Specifically, we provide a generic, practical generalization bound and an associated learning principle that apply universally to a large family of IW regularizations, enabling a fair comparison in practice on OPL tasks.

This paper is organized as follows. Section 2 provides the necessary background on IPS estimators and IW regularizations. Section 3 reviews related work, focusing on the guarantees and learning principles found in the OPL literature. Section 4 presents our PAC-Bayesian generalization bounds for regularized IPS and introduces our learning principles derived from these bounds. Finally, Section 5 compares different IW regularizations on real-world datasets.

## 2 BACKGROUND

### 2.1 OFFLINE CONTEXTUAL BANDITS

An agent interacts with a *contextual bandit* environment over $n$ rounds. In round $i \in [n]$, the agent observes a *context* $x_i \sim \nu$, where $\nu$ is a distribution with support $\mathcal{X} \subseteq \mathbb{R}^d$, a $d$-dimensional *compact context space*. The agent then selects an *action* $a_i$ from a *finite action space* $\mathcal{A} = [K]$. This action is sampled as $a_i \sim \pi_0(\cdot|x_i)$, where $\pi_0$ is the logging policy used to collect data. Specifically, for a given context $x$, $\pi_0(a|x)$ represents the probability that the agent takes action $a$ under its current (logging) policy. Finally, the agent receives a stochastic cost[1] $c_i \in [-1, 0]$ that depends on the observed context $x_i$ and the action $a_i$. Precisely, $c_i \sim p(\cdot|x_i, a_i)$, where $p(\cdot|x, a)$ is the *cost distribution* of action

---

[1]For simplicity, we assume that costs $c \in [-1, 0]$, though this can be easily extended to $c \in [-C, 0]$ for $C > 0$.

$a$ in context $x$. The expected cost of action $a$ in context $x$ is given by the *cost function* $c(x, a) = \mathbb{E}_{c \sim p(\cdot|x,a)}[c]$. Using an alternative terminology, costs can be defined as the negative of rewards: for any $(x, a) \in \mathcal{X} \times \mathcal{A}, c(x, a) = -r(x, a)$, where $r : \mathcal{X} \times \mathcal{A} \to [0, 1]$ is the *reward function*. These interactions result in an $n$-sized logged data $S = (x_i, a_i, c_i)_{i \in [n]}$, where $(x_i, a_i, c_i)$ are i.i.d from $\mu_\pi$, the joint distribution of $(x, a, c)$ defined as $\mu_\pi(x, a, c) = \nu(x)\pi(a|x)p(c|x, a)$ for any $(x, a, c) \in \mathcal{X} \times \mathcal{A} \times [-1, 0]$.

Agents are represented by stochastic policies $\pi \in \Pi$, where $\Pi$ denotes the space of policies. Specifically, for a given context $x \in \mathcal{X}$, $\pi(\cdot|x)$ defines a probability distribution over the action space $\mathcal{A}$. Then, the performance of a policy $\pi \in \Pi$ is measured by the *risk*, defined as

$$R(\pi) = \mathbb{E}_{x \sim \nu, a \sim \pi(\cdot|x)}[c(x, a)] . \tag{1}$$

Given a policy $\pi \in \Pi$ and logged data $S$, the goal of OPE is to design an estimator $\hat{R}(\pi, S)$ for the true risk $R(\pi)$ such that $\hat{R}(\pi, S) \approx R(\pi)$. Leveraging this estimator, OPL aims to find a policy $\hat{\pi}_n \in \Pi$ such that $R(\hat{\pi}_n) \approx \min_{\pi \in \Pi} R(\pi)$. We focus on the IPS estimator [Horvitz and Thompson, 1952], which estimates the risk $R(\pi)$ by re-weighting samples using the ratio between $\pi$ and $\pi_0$

$$\hat{R}_{\text{IPS}}(\pi, S) = \frac{1}{n} \sum_{i=1}^{n} w(x_i, a_i)c_i , \tag{2}$$

where for any $(x, a) \in \mathcal{X} \times \mathcal{A}$, $w(x, a) = \pi(a|x)/\pi_0(a|x)$ are the *importance weights (IWs)*.

### 2.2 REGULARIZED IMPORTANCE WEIGHTING

The IPS estimator in (2) is unbiased when $\pi_0(a|x) = 0$ implies that $\pi(a|x) = 0$ for all $(x, a) \in \mathcal{X} \times \mathcal{A}$. However, its variance scales linearly with the IWs [Swaminathan et al., 2017] and can be large if the target policy $\pi$ differs significantly from the logging policy $\pi_0$. To mitigate this effect, it is common to transform the IWs using a regularization function that introduces some bias to reduce variance. Specifically, a regularized IPS estimator is defined as

$$\hat{R}(\pi, S) = \frac{1}{n} \sum_{i=1}^{n} \hat{w}(x_i, a_i)c_i , \tag{3}$$

where $\hat{w}(x, a)$ are the regularized IWs. Examples of $\hat{w}$ include clipping (Clip) [London and Sandler, 2019], exponential smoothing (ES) [Aouali et al., 2023a], implicit exploration (IX) [Gabbianelli et al., 2024], and harmonic (Har) [Metelli et al., 2021], defined as

$$\text{Clip:} \quad \hat{w}(x, a) = \frac{\pi(a|x)}{\max(\pi_0(a|x), \tau)} , \ \tau \in [0, 1] , \quad (4)$$

$$\text{ES:} \quad \hat{w}(x, a) = \frac{\pi(a|x)}{\pi_0(a|x)^\alpha} , \ \alpha \in [0, 1] ,$$

$$\text{IX:} \quad \hat{w}(x, a) = \frac{\pi(a|x)}{\pi_0(a|x) + \gamma} , \ \gamma \in [0, 1] ,$$

$$\text{Har:} \quad \hat{w}(x, a) = \frac{w(x, a)}{(1 - \lambda)w(x, a) + \lambda} , \ \lambda \in [0, 1] .$$

These regularizations are linear in $\pi$ except `Har`. Other nonlinear regularizations have been proposed [Swaminathan and Joachims, 2015a, Su et al., 2020], but we will focus on the above examples because their hyperparameters fall within the same range $[0, 1]$, facilitating their comparison.

# 3 PESSIMISTIC LEARNING PRINCIPLES

We now discuss the theoretical guarantees and pessimistic learning principles previously derived in the literature. An extended related work section can be found in Appendix A.

Let $\hat{R}$ be an estimator of the risk $R$. In OPL, the goal is to minimize the unknown risk $R$ using the estimator $\hat{R}$. Pessimistic learning principles typically penalize $\hat{R}$, aiming to find $\hat{\pi}_n = \operatorname{argmin}_{\pi \in \Pi} \hat{R}(\pi, S) + \operatorname{pen}(\pi, S)$, with the expectation that $R(\hat{\pi}_n) \approx \min_{\pi \in \Pi} R(\pi)$. The penalization term $\operatorname{pen}(\cdot, S)$ is derived using one of the following methods.

**The Use of Evaluation Bounds.** Metelli et al. [2021] derived *evaluation* bounds for the `Har` regularization in (4) and used them to formulate a pessimistic OPL learning principle. Specifically, they showed that the following inequality holds for a *fixed target policy* $\pi \in \Pi$ and $\delta \in (0, 1)$

$$\mathbb{P}\big(\big|R(\pi) - \hat{R}(\pi, S)\big| \le f(\delta, \pi, \pi_0, n)\big) \ge 1 - \delta, \quad (5)$$

for some function $f$. Essentially, (5) indicates that for a fixed policy $\pi \in \Pi$, the event $|R(\pi) - \hat{R}(\pi, S)| \le f(\delta, \pi, \pi_0, n)$ holds with high probability. However, this event depends on the target policy $\pi$. Thus (5) is useful for evaluating a *single target policy* when having access to *multiple logged data sets* $S$. This poses a problem for OPL, where we optimize over a potentially *infinite space of policies* using a *single logged data set* $S$. This is the fundamental theoretical limitation of using evaluation bounds similar to (5) in OPL. While it is possible to transform (5) into a generalization bound that simultaneously holds for any policy $\pi \in \Pi$ by applying a union bound, this approach may result in intractable complexity terms and, consequently, intractable pessimistic learning principles.

**The Use of One-Sided Generalization Bounds.** Alternatively, generalization bounds [Swaminathan and Joachims, 2015a, London and Sandler, 2019, Sakhi et al., 2023] address the limitations of evaluation bounds. These bounds generally take the following form: for $\delta \in (0, 1)$,

$$\mathbb{P}\big(\forall \pi \in \Pi, R(\pi) \le \hat{R}(\pi, S) + f(\delta, \Pi, \pi, \pi_0, n)\big) \quad (6)$$
$$\ge 1 - \delta,$$

where the function $f$ now depends on the space of policies $\Pi$. The key difference between (5) and (6) is that here the event $R(\pi) \le \hat{R}(\pi, S_\Pi) + f(\delta, \Pi, \pi, \pi_0, n)$ holds with high probability for all target policies $\pi$. Since this is a high-probability event, we assume it holds for our logged data $S$.

This is then used to define the learned policy $\hat{\pi}_n \in \Pi$ as

$$\hat{\pi}_n = \operatorname*{argmin}_{\pi \in \Pi} \hat{R}(\pi, S) + f(\delta, \Pi, \pi, \pi_0, n). \quad (7)$$

The issue with (6) is that it is a *one-sided* inequality that does not attest to the quality of the estimator $\hat{R}$. To illustrate, consider that with probability 1, $R(\pi) \le \hat{R}^{\text{POOR}}(\pi)$, using a poor estimator of the risk, $\hat{R}^{\text{POOR}}(\pi) = 0$ for any $\pi \in \Pi$. This holds because, by definition, $R(\pi) \in [-1, 0]$. However, $\hat{R}^{\text{POOR}}$ is not informative about $R$, making its minimization irrelevant. Thus we need to control the quality of the upper bound on $R$, which is achieved by *two-sided* inequalities

$$\mathbb{P}\big(\forall \pi \in \Pi, |R(\pi) - \hat{R}(\pi, S)| \le f(\delta, \Pi, \pi, \pi_0, n)\big) \quad (8)$$
$$\ge 1 - \delta.$$

Here, the pessimistic learning principle in (7) uses the function $f$ from the two-sided inequality in (8). In particular, this allows us to derive high-probability inequalities on the suboptimality (SO) gap of $\hat{\pi}_n$, which is the difference $R(\hat{\pi}_n) - R(\pi_*)$. Specifically, we can show that $R(\hat{\pi}_n) - R(\pi_*) \le 2f(\delta, \Pi, \pi_*, \pi_0, n)$, where $\hat{\pi}_n$ is the learned policy from (7) (with $f$ obtained from the two-sided inequality in (8)) and $\pi_* = \operatorname{argmin}_{\pi \in \Pi} R(\pi)$ is the optimal policy. This demonstrates why pessimism is appealing in OPL: the suboptimality gap of the learned policy $\hat{\pi}_n$, i.e., $R(\hat{\pi}_n) - R(\pi_*)$, is bounded by $2f(\delta, \Pi, \pi_*, \pi_0, n)$, where $f$ is evaluated at the optimal policy $\pi_*$. Consequently, the risk estimator $\hat{R}$ needs to be precise only for the optimal policy, rather than for all policies within the class $\Pi$.

**The Use of Heuristics.** Many studies have proposed specific heuristics where a simplified function $g$ is used instead of the theoretical function $f$ in (7). For example, Swaminathan and Joachims [2015a] minimized the estimated risk while penalizing the empirical variance of the estimator. This approach was inspired by a generalization bound with a function $f$ that includes a variance term but discards more complicated terms from the bound, such as the covering number of the policy space $\Pi$. Similarly, London and Sandler [2019] parameterized policies by a mean parameter and proposed penalizing the estimated risk by the $L_2$ distance between the means of the logging and target policies, discarding all other terms from their generalization bound. While these heuristics lead to tractable and computationally attractive objectives, they often lack theoretical justification and guarantees. We note that pessimistic principles have been used in a different context than regularized IPS estimators. For example, Wang et al. [2024] proposed a heuristic approach where the standard (non-regularized) IPS estimator $\hat{R}_{\text{IPS}}(\pi, S)$ in (2) is penalized with a pseudo-loss $\text{PL}(\pi, S) = \frac{1}{n} \sum_{i \in [n]} \sum_{a \in \mathcal{A}} \frac{\pi(a|x_i)}{\pi_0(a|x_i)}$. Precisely, they defined $\hat{\pi}_n = \operatorname{argmin}_{\pi \in \Pi} \hat{R}_{\text{IPS}}(\pi, S) + \beta \text{PL}(\pi, S)$, where $\beta$ is a hyperparameter. They upper bounded the suboptimality gap of their $\hat{\pi}_n$ for a specific theoretical choice of $\beta$.

**The Use of Implicit Pessimism.** Recently, Gabbianelli et al. [2024] proposed the use of the `IX`-estimator in (4) in OPL

and demonstrated that, with careful analysis, they could obtain tight bounds. They observed that the IX-estimator exhibits asymmetry and thus did not use a single two-sided inequality to derive their bound. Instead, they analyzed each side individually using distinct methods and combined the results to obtain the desired two-sided inequality. In particular, this allowed them to derive an upper bound function $f$ that depends only on the policy space $\Pi$, confidence level $\delta$, and the number of samples $n$, such that $f(\delta, \Pi, \pi, \pi_0, n) = f(\delta, \Pi, n)$. This led them to define

$$
\begin{aligned}
\hat{\pi}_n &= \underset{\pi \in \Pi}{\operatorname{argmin}} \, \hat{R}(\pi, S) + f(\delta, \Pi, \pi, \pi_0, n) \,, \qquad (9) \\
&= \underset{\pi \in \Pi}{\operatorname{argmin}} \, \hat{R}(\pi, S) + f(\delta, \Pi, n) = \underset{\pi \in \Pi}{\operatorname{argmin}} \, \hat{R}(\pi, S) \,,
\end{aligned}
$$

where the principle of pessimism becomes equivalent to directly minimizing the estimator since $f$ does not depend on $\pi$. This approach is appealing as it avoids computing potentially heavy statistics of the upper bound while still enjoying the benefits of pessimism. However, it requires a careful analysis of the specific IW regularization, whereas we provide a generic bound that holds for any IW regularization. Following Aouali et al. [2023a], we directly derive two-sided bounds for regularized IPS, which might be loose depending on the logging policy (Appendix C.5) but still lead to good empirical performance (Section 5). Investigating similar asymmetric analysis in general regularized IPS is an interesting avenue for future work.

**Our Approach.** We derive a two-sided generalization bound that holds simultaneously for any policy $\pi \in \Pi$, as outlined in (8). We examine two pessimistic learning principles: directly optimizing the bound or optimizing a simplified penalty inspired by it (heuristic). Both principles apply to any IW regularization, including the standard, non-regularized IPS. Our theory builds on the proof of Pac-Bayesian bounds in Aouali et al. [2023a], extending its scope beyond the ES regularization in (4) to include other IW regularizations. A limitation of the previous work was the empirical comparison of different pessimistic learning principles, each employing a different IW regularization for IPS. For example, Aouali et al. [2023a] compared optimizing ES-IPS penalized by their generalization bound with optimizing Clip-IPS penalized by existing bounds (e.g., [Sakhi et al., 2023, London and Sandler, 2019]). Although they demonstrated significant improvements in OPL performance with ES, they did not determine whether these improvements were due to the new IW regularization technique (ES vs. Clip) or the new generalization bound (their bounds vs. those in Sakhi et al. [2023], London and Sandler [2019]). This ambiguity motivates our development of a generic generalization bound that applies universally to any IW regularization and also serves as the basis for a generic heuristic inspired by it.

## 4 THEORETICAL ANALYSIS

We derive our PAC-Bayes generalization bound for the regularized IPS estimator $\hat{R}(\pi)$ in (3) under the assumption that $\hat{w}(x, a) = g(\pi(a|x), \pi_0(a|x))$ for any $(x, a) \in \mathcal{X} \times \mathcal{A}$, where $g : [0, 1] \times [0, 1] \to \mathbb{R}^+$. This assumption is broadly applicable and aligns with known IW regularizations. We make it to explicitly clarify the dependence on $\pi(a|x)$ and purposefully exclude self-normalized IW, where $\hat{w}(x_i, a_i) = n w(x_i, a_i) / \sum_{j \in [n]} w(x_j, a_j)$. In self-normalized IW, the regularization depends not only on the specific pair $x_i, a_i$ but also on all other pairs $x_j, a_j$, which is not supported by our theory.

### 4.1 INTRODUCTION TO PAC-BAYES THEORY

Consider learning problems specified by an instance space denoted as $\mathcal{Z}$, a hypothesis space $\mathcal{H}$ consisting of predictors $h$, and a loss function $L : \mathcal{H} \times \mathcal{Z} \to \mathbb{R}$. Assume access to a dataset $S = (z_i)_{i \in [n]}$, where $z_1, \ldots, z_n$ are i.i.d. from an unknown distribution $\mathbb{D}$. The risk of a hypothesis $h$ is defined as $R(h) = \mathbb{E}_{z \sim \mathbb{D}}[L(h, z)]$, while its empirical counterpart is denoted as $\hat{R}(h, S) = \frac{1}{n} \sum_{i=1}^{n} L(h, z_i)$.

In PAC-Bayes, our primary focus is to examine the average generalization capabilities under a distribution $\mathbb{Q}$ on $\mathcal{H}$ by controlling the difference between the expected risk under $\mathbb{Q}$ (expressed as $\mathbb{E}_{h \sim \mathbb{Q}}[R(h)]$) and the expected empirical risk under $\mathbb{Q}$ (expressed as $\mathbb{E}_{h \sim \mathbb{Q}}[\hat{R}(h, S)]$).

An example of PAC-Bayes generalization bounds originally proposed by McAllester [1998] is as follows. Assume that the values of $L(h, z) \in [0, 1]$ for any $(h, z) \in \mathcal{H} \times \mathcal{Z}$, and that we have a fixed prior distribution $\mathbb{P}$ on $\mathcal{H}$ and a parameter $\delta$ that falls within $(0, 1)$. Then, with a probability of at least $1 - \delta$ over the sample set $S$ drawn from $\mathbb{D}^n$, it holds simultaneously for any distribution $\mathbb{Q}$ on $\mathcal{H}$ that

$$
\mathbb{E}_{h \sim \mathbb{Q}}[R(h)] \leq \mathbb{E}_{h \sim \mathbb{Q}}[\hat{R}(h, S)] + \sqrt{\frac{D_{\mathrm{KL}}(\mathbb{Q} \| \mathbb{P}) + \log \frac{2\sqrt{n}}{\delta}}{2n}} \,,
$$

where $D_{\mathrm{KL}}$ denotes the Kullback-Leibler divergence. The reader may refer to Alquier [2021] for a comprehensive introduction to PAC-Bayes theory.

### 4.2 GENERALIZATION BOUNDS FOR OPL

Let $d'$ be a positive integer, and let $\Theta \subset \mathbb{R}^{d'}$ be a $d'$-dimensional parameter space. We parametrize our learning policies as $\pi_\theta$, defining our space of policies as $\Pi = \{\pi_\theta; \theta \in \Theta\}$. An example of this is the softmax policy, parameterized as follows

$$
\pi_\theta^{\mathrm{SOF}}(a|x) = \frac{\exp(\phi(x)^\top \theta_a)}{\sum_{a' \in \mathcal{A}} \exp(\phi(x)^\top \theta_{a'})} \,, \qquad (10)
$$

where $\theta_a \in \mathbb{R}^d$ and consequently $\theta = (\theta_a)_{a \in \mathcal{A}} \in \mathbb{R}^{dK}$, with $d' = dK$. Moreover, let $\mathbb{Q}$ be a distribution on the

parameter space $\Theta$. Then PAC-Bayes theory allows us to control the quantity $\left|\mathbb{E}_{\theta\sim\mathbb{Q}}[R(\pi_\theta) - \hat{R}(\pi_\theta, S)]\right|$, where

$$R(\pi_\theta) = \mathbb{E}_{x\sim\nu, a\sim\pi_\theta(\cdot|x)}[c(x,a)],$$

$$\hat{R}(\pi_\theta, S) = \frac{1}{n}\sum_{i=1}^{n}\hat{w}_\theta(x_i, a_i)c_i,$$

with $\hat{w}_\theta(x,a) = g(\pi_\theta(a|x), \pi_0(a|x))$. We also assume that the costs are deterministic for ease of exposition. The same result holds for stochastic costs. The proof is provided in Appendix B.

**Theorem 1.** *Let $\lambda > 0$, $n \geq 1$, $\delta \in (0,1)$, and let $\mathbb{P}$ be a fixed prior on $\Theta$. The following inequality holds with probability at least $1 - \delta$ for any distribution $\mathbb{Q}$ on $\Theta$:*

$$\left|\mathbb{E}_{\theta\sim\mathbb{Q}}[R(\pi_\theta) - \hat{R}(\pi_\theta, S)]\right| \quad (11)$$

$$\leq \sqrt{\frac{\mathrm{KL}_1(\mathbb{Q})}{2n}} + \frac{\mathrm{KL}_2(\mathbb{Q})}{n\lambda} + B_n(\mathbb{Q}) + \frac{\lambda}{2}\bar{V}_n(\mathbb{Q}),$$

*where* $\mathrm{KL}_1(\mathbb{Q}) = D_{\mathrm{KL}}(\mathbb{Q}\|\mathbb{P}) + \log\frac{4\sqrt{n}}{\delta}$, $\mathrm{KL}_2(\mathbb{Q}) = D_{\mathrm{KL}}(\mathbb{Q}\|\mathbb{P}) + \log\frac{4}{\delta}$, *and*

$$\bar{V}_n(\mathbb{Q}) = \frac{1}{n}\sum_{i=1}^{n}\mathbb{E}_{\theta\sim\mathbb{Q}}\big[\mathbb{E}_{a\sim\pi_0(\cdot|x_i)}[\hat{w}_\theta(x_i,a)^2]$$
$$+ \hat{w}_\theta(x_i,a_i)^2 c_i^2\big],$$

*and* $B_n(\mathbb{Q}) = \frac{1}{n}\sum_{i=1}^{n}\sum_{a\in\mathcal{A}}\mathbb{E}_{\theta\sim\mathbb{Q}}\big[|\pi_\theta(a|x_i)$
$$- \pi_0(a|x_i)\hat{w}_\theta(x_i,a)|\big].$$

Generally, the bound is tractable due to the conditioning on the contexts $(x_i)_{i\in[n]}$, allowing us to bypass the need for computing the unknown expectation $\mathbb{E}_{x\sim\nu}[\cdot]$. Recall that the prior $\mathbb{P}$ is any fixed distribution over $\Theta$. In particular, if a $\theta_0$ exists such that the logging policy $\pi_0 = \pi_{\theta_0}$, then $\mathbb{P}$ can be specified as Gaussian with mean $\theta_0$ and some covariance. The terms $\mathrm{KL}_1(\mathbb{Q})$ and $\mathrm{KL}_2(\mathbb{Q})$ contain the divergence $D_{\mathrm{KL}}(\mathbb{Q}\|\mathbb{P})$, which penalizes posteriors $\mathbb{Q}$ that deviate significantly from the prior $\mathbb{P}$. The latter can be computed in closed-form if both $\mathbb{P}, \mathbb{Q}$ are Gaussian for instance. Moreover, $B_n(\mathbb{Q})$ represents the bias introduced by the IW regularization, given contexts $(x_i)_{i\in[n]}$; $B_n(\mathbb{Q}) = 0$ when $\hat{w}_\theta(x,a) = w(x,a)$ (no IW regularization) and $B_n(\mathbb{Q}) > 0$ otherwise. The first term in $\bar{V}_n(\mathbb{Q})$ resembles the theoretical second moment of the regularized IWs $\hat{w}_\theta(x,a)$ (without the cost) when viewed as random variables, while the second term resembles the empirical second moment of $\hat{w}_\theta(x,a)c$ (with the cost). If $\bar{V}_n(\mathbb{Q})$ is bounded (which is the case for all IW regularizations in Section 2.2 except ES), we can set $\lambda = 1/\sqrt{n}$, resulting in a $\mathcal{O}(1/\sqrt{n} + B_n(\mathbb{Q}))$ bound.

**Linear vs. Non-linear IW Regularization.** If $\hat{w}(x,a)$ is linear in $\pi_\theta(x,a)$ (i.e., $g$ linear in its first variable), then $\hat{R}$

is also linear in $\pi_\theta$, yielding

$$\left|\mathbb{E}_{\theta\sim\mathbb{Q}}[R(\pi_\theta) - \hat{R}(\pi_\theta, S)]\right| = \left|R(\pi_\mathbb{Q}) - \hat{R}(\pi_\mathbb{Q}, S)\right|,$$

where we define

$$\pi_\mathbb{Q} = \mathbb{E}_{\theta\sim\mathbb{Q}}[\pi_\theta]. \quad (12)$$

This technique is widely used in the literature [London and Sandler, 2019, Sakhi et al., 2023, Aouali et al., 2023a] because it allows translating the bound in Theorem 1, which controls $\left|\mathbb{E}_{\theta\sim\mathbb{Q}}[R(\pi_\theta) - \hat{R}(\pi_\theta, S)]\right|$, into a bound that controls $|R(\pi_\mathbb{Q}) - \hat{R}(\pi_\mathbb{Q}, S)|$, the quantity of interest in OPL. The main requirement is to find linear IW regularizations and policies that satisfy (12). Fortunately, many IW regularizations, such as Clip, IX, and ES in (4), are linear in $\pi$, and several practical policies adhere to the formulation in (12); refer to Aouali et al. [2023a, Section 4.2] for an in-depth explanation of such policies, including softmax, mixed-logit, and Gaussian policies. In fact, Sakhi et al. [2023] demonstrated that any policy can be written as (12).

In Corollary 2, we specialize Theorem 1 under linear IW regularizations of the form $\hat{w}_\theta(x,a) = \frac{\pi_\theta(a|x)}{h(\pi_0(a|x))}$, assuming $h(\pi_0(a|x)) \geq \pi_0(a|x)$ for any $(x,a) \in \mathcal{X} \times \mathcal{A}$. Additionally, we assume that $\pi_\theta$ is binary, meaning $\pi_\theta(a \mid x) \in \{0,1\}$ for any $(x,a) \in \mathcal{X} \times \mathcal{A}$. In other words, $\pi_\theta$ is deterministic, allowing us to use $\pi_\theta(a|x)^2 = \pi_\theta(a|x)$ for any $(x,a) \in \mathcal{X} \times \mathcal{A}$. Essentially, the policies $\pi_\mathbb{Q}$ defined in (12) can be viewed as a mixture of deterministic policies under $\mathbb{Q}$. Note that this assumption on $\pi_\theta$ being binary is mild. For instance, policies $\pi_\mathbb{Q}$ that can be written as mixtures of binary $pi_\theta$ include softmax, mixed-logit, and Gaussian policies [Aouali et al., 2023a, Section 4.2]. Under these assumptions, Theorem 1 yields the following result.

**Corollary 2.** *Assume the regularized IWs can be written as $\hat{w}_\theta(x,a) = \frac{\pi_\theta(a|x)}{h(\pi_0(a|x))}$ with $h : [0,1] \to \mathbb{R}^+$ verifies $h(p) \geq p$ for any $p \in [0,1]$. Moreover, for any distribution $\mathbb{Q}$ in the parameter space $\Theta$, we define $\pi_\mathbb{Q} = \mathbb{E}_{\theta\sim\mathbb{Q}}[\pi_\theta]$ where $\pi_\theta$ is binary. Then, let $\lambda > 0$, $n \geq 1$, $\delta \in (0,1)$, and let $\mathbb{P}$ be a fixed prior on $\Theta$, The following inequality holds with probability at least $1 - \delta$ for any distribution $\mathbb{Q}$ on $\Theta$*

$$\left|R(\pi_\mathbb{Q}) - \hat{R}(\pi_\mathbb{Q}, S)\right| \quad (13)$$

$$\leq \sqrt{\frac{\mathrm{KL}_1(\mathbb{Q})}{2n}} + B_n(\pi_\mathbb{Q}) + \frac{\mathrm{KL}_2(\mathbb{Q})}{n\lambda} + \frac{\lambda}{2}\bar{V}_n(\pi_\mathbb{Q}),$$

*where* $\mathrm{KL}_1(\mathbb{Q})$ *and* $\mathrm{KL}_2(\mathbb{Q})$ *are defined in Theorem 1, and*

$$\bar{V}_n(\pi_\mathbb{Q}) = \frac{1}{n}\sum_{i=1}^{n}\mathbb{E}_{a\sim\pi_0(\cdot|x_i)}\left[\frac{\pi_\mathbb{Q}(a|x_i)}{h(\pi_0(a|x_i)^2}\right]$$
$$+ \frac{\pi_\mathbb{Q}(a_i|x_i)}{h(\pi_0(a_i|x_i))^2}c_i^2,$$

$$B_n(\pi_\mathbb{Q}) = 1 - \frac{1}{n}\sum_{i=1}^{n}\sum_{a\in\mathcal{A}}\pi_0(a|x_i)\frac{\pi_\mathbb{Q}(a|x_i)}{h(\pi_0(a|x_i))}.$$

The terms in the above bound have similar interpretations to those in Theorem 1. The main benefit of Corollary 2 is that it eliminates the need for the expectation $\mathbb{E}_{\theta \sim \mathbb{Q}}[\cdot]$, which is now embedded in the definition of policies in (12). For example, Corollary 2 allows us to recover the main result of ES in Aouali et al. [2023a] when $h(p) = p^\alpha$, $\alpha \in [0, 1]$. Similarly, we can apply it to IX [Gabbianelli et al., 2024] by setting $h(p) = p + \gamma$, $\gamma \geq 0$, and to Clip [London and Sandler, 2019] by setting $h(p) = \max(p, \tau)$, $\tau \in [0, 1]$.

Finally, if $\hat{w}_\theta(x, a)$ is not linear in $\pi_\theta$, then this technique cannot be used, and the original expectation $\mathbb{E}_{\theta \sim \mathbb{Q}}[\cdot]$ in Theorem 1 must be retained.

**Limitations.** This bound has two main limitations. **1)** Despite its broad applicability, directly applying Theorem 1 to bound the suboptimality gap of the learned policy-specifically, to bound $R(\hat{\pi}_n) - R(\pi_*)$, where $\pi_* = \arg\min_{\pi \in \Pi} R(\pi)$ is the optimal policy and $\hat{\pi}_n$ is learned by optimizing the bound-is not straightforward. To illustrate, consider the linear IW regularization case in Corollary 2 and suppose that $\pi_*$ can be expressed as $\pi_* = \pi_{\mathbb{Q}_*}$, where $\mathbb{Q}_* = \arg\min_{\mathbb{Q}} R(\pi_{\mathbb{Q}})$ is the optimal distribution. In this scenario, the suboptimality gap would be bounded by the upper bound in Corollary 2, evaluated at the optimal distribution $\mathbb{Q} = \mathbb{Q}_*$. However, the scaling of this suboptimality bound with $n$ is not immediately evident for general IW regularizers and requires individual examination for each IW regularization. This is because the bound contains numerous empirical (data-dependent) terms such as $B_n(\pi_{\mathbb{Q}_*})$ and $\bar{V}_n(\pi_{\mathbb{Q}_*})$ that are not easily transformed into data-independent terms that scale as $\mathcal{O}(1/\sqrt{n})$. Nonetheless, the versatility, tractability, and proven empirical benefits of our bound (Section 5) make it appealing. **2)** It has been noted that directly deriving two-sided bounds for IW estimators might be loose because they treat both tails similarly, whereas prior work [Gabbianelli et al., 2024] indicates essential differences between the lower and upper tails, as seen in the IX-estimator [Gabbianelli et al., 2024]. This work directly derives two-sided bounds for general regularized IPS. Investigating whether bounding each side individually could lead to terms that are easier to interpret and solve the above problem is an interesting direction for future research.

### 4.3 PESSIMISTIC LEARNING PRINCIPLES

Theorem 1 yields two pessimistic learning principles.

**Bound Optimization.** First, one can directly learn a $\hat{\pi}_n$ that optimizes the bound of Theorem 1 as follows

$$\underset{\mathbb{Q}}{\arg\max} \, \mathbb{E}_{\theta \sim \mathbb{Q}} \left[ \hat{R}(\pi_\theta, S) \right] + \sqrt{\frac{\mathrm{KL}_1(\mathbb{Q})}{2n}} + B_n(\mathbb{Q})$$
$$+ \frac{\mathrm{KL}_2(\mathbb{Q})}{n\lambda} + \frac{\lambda}{2} \bar{V}_n(\mathbb{Q}), \quad (14)$$

Here, the main challenge is that the objective involves an ex-

pectation under $\mathbb{Q}$. Fortunately, the reparameterization trick [Kingma et al., 2015] can be used in this case. This trick allows us to express a gradient of an expectation as an expectation of a gradient, which can then be estimated using the empirical mean (Monte Carlo approximation). In our case, we use the *local* reparameterization trick [Kingma et al., 2015], known for reducing the variance of stochastic gradients. Specifically, we consider softmax policies $\pi_\theta^{\mathrm{SOF}}(a|x)$ in (10) and set $\mathbb{Q} = \mathcal{N}(\mu, \sigma^2 I_{dK})$ where $\mu \in \mathbb{R}^{dK}$ and $\sigma > 0$ are learnable parameters. Then, all terms in (14) are of the form $\mathbb{E}_{\theta \sim \mathcal{N}(\mu, \sigma^2 I_{dK})}[f(\pi_\theta^{\mathrm{SOF}}(a|x))]$ for some function $f$. These terms can be rewritten as

$$\mathbb{E}_{\theta \sim \mathcal{N}(\mu, \sigma^2 I_{dK})}[f(\pi_\theta^{\mathrm{SOF}}(a|x))]$$
$$= \mathbb{E}_{\epsilon \sim \mathcal{N}(0, \|\phi(x)\|_2^2 I_K)} \left[ f\left( \frac{\exp(\phi(x)^\top \mu_a + \sigma \epsilon_a)}{\sum_{a' \in \mathcal{A}} \exp(\phi(x)^\top \mu_{a'} + \sigma \epsilon_{a'})} \right) \right].$$

This expectation is approximated by generating i.i.d. samples $\epsilon_i \sim \mathcal{N}(0, \|\phi(x)\|_2^2 I_K)$ and computing the corresponding empirical mean. The gradients are approximated similarly (Appendix C.2). Unfortunately, these techniques can induce high variance when the number of actions $K$ is large. This can be mitigated by considering linear IW regularizations and optimizing the bound in Corollary 2. However, this technique only works for linear IW regularizations. Therefore, we propose another practical learning principle inspired by our bound in Theorem 1, which enhances performance at the cost of additional hyperparameters.

**Heuristic Optimization.** The following heuristic avoids the obstacles of directly optimizing the bound, at the cost of introducing some hyperparameters, while still being inspired by Theorem 1. This approach involves minimizing the estimated risk $\hat{R}$, penalized by its associated bias and variance terms from Theorem 1, along with a proximity term to the logging policy $\pi_0 = \pi_{\theta_0}$ such as

$$\hat{R}(\pi_\theta, S) + \lambda_1 \|\theta - \theta_0\|^2 + \lambda_2 \tilde{V}_n(\pi_\theta) + \lambda_3 \tilde{B}_n(\pi_\theta), \quad (15)$$

where $\tilde{V}_n(\pi_\theta)$ and $\tilde{B}_n(\pi_\theta)$ are the terms inside the expectations in $\bar{V}_n(\mathbb{Q})$ and $B_n(\mathbb{Q})$, respectively, $\theta_0$ is the parameter of $\pi_0$, and $\lambda_1, \lambda_2, \lambda_3$ are tunable hyperparameters.

Both learning principles in (14) and (15) are suitable for stochastic gradient descent. They are also generic, enabling the comparison of different IW regularization techniques given a fixed, observed logged data $S$. In Section 5, we will empirically compare these two learning principles and evaluate the effect of different IW regularization techniques.

### 4.4 SKETCH OF PROOF FOR THE MAIN RESULT

Our goal is to bound $\mathbb{E}_{\theta \sim \mathbb{Q}}\left[ R(\pi_\theta) - \hat{R}(\pi_\theta, S) \right]$. To achieve this, we decompose it into three terms as follows

$$\mathbb{E}_{\theta \sim \mathbb{Q}}[R(\pi_\theta) - \hat{R}(\pi_\theta, S)] = I_1 + I_2 + I_3,$$

Next, we explain the terms $I_1$, $I_2$, and $I_3$ and the rationale for their introduction.

First, $I_1 = \mathbb{E}_{\theta \sim \mathbb{Q}}\left[R(\pi_\theta) - \frac{1}{n}\sum_{i=1}^n R(\pi_\theta|x_i)\right]$, where $R(\pi_\theta|x_i) = \mathbb{E}_{a \sim \pi_\theta(\cdot|x_i)}[c(x_i,a)]$, represents the risk given context $x_i$. This term captures the estimation error of the empirical mean of the risk using $n$ i.i.d. contexts $(x_i)_{i \in [n]}$. It is introduced to avoid the intractable expectation over $x \sim \nu$, thereby leading to a tractable bound that can be directly used in our pessimistic learning principle.

Second, $I_2 = \frac{1}{n}\sum_{i=1}^n \mathbb{E}_{\theta \sim \mathbb{Q}}\left[R(\pi_\theta|x_i) - \hat{R}(\pi_\theta|x_i)\right]$, with $\hat{R}(\pi_\theta|x_i) = \mathbb{E}_{a \sim \pi_0(\cdot|x_i)}[\hat{w}_\theta(x_i,a)c(x_i,a)]$, represents the expectation of the risk estimator given $x_i$. This term is a bias term conditioned on the contexts $(x_i)_{i \in [n]}$, and its absolute value can be bounded by tractable terms.

Finally, $I_3 = \frac{1}{n}\sum_{i=1}^n \mathbb{E}_{\theta \sim \mathbb{Q}}\left[\hat{R}(\pi_\theta|x_i) - \hat{R}(\pi_\theta, S)\right]$ represents the estimation error of the risk conditioned on the contexts $(x_i)_{i \in [n]}$. This conditioning allows us to avoid the unknown expectation over $x \sim \nu$, making it possible to bound $|I_3|$ by tractable terms.

These terms are bounded as follows. **1)** Alquier [2021, Theorem 3.3] allows bounding $I_1$ with high probability as $|I_1| \leq \sqrt{\frac{\mathsf{KL}_1(\mathbb{Q})}{2n}}$. **2)** Using the fact that $|c(x,a)| \leq 1$ for any $(x,a) \in \mathcal{X} \times \mathcal{A}$, $|I_2|$ can be bounded as $|I_2| \leq B_n(\mathbb{Q})$. **3)** Bounding $|I_3|$ is more challenging. We manage this by expressing the term using martingale difference sequences and adapting [Haddouche and Guedj, 2022, Theorem 2.1]. Let $(\mathcal{F}_i)_{i \in \{0\} \cup [n]}$ be a filtration adapted to $(S_i)_{i \in [n]}$ where $S_i = (a_\ell)_{\ell \in [i]}$ for any $i \in [n]$. Then define

$$f_i(a_i, \pi_\theta) = \mathbb{E}_{a \sim \pi_0(\cdot|x_i)}[\hat{w}_\theta(x_i,a)c(x_i,a)] \\ - \hat{w}_\theta(x_i,a_i)c(x_i,a_i).$$

We show that for any $\theta \in \Theta$, $(f_i(a_i,\pi_\theta))_{i \in [n]}$ forms a martingale difference sequence, which yields

$$|\mathbb{E}_{\theta \sim \mathbb{Q}}[M_n(\theta)]| \leq \frac{\mathsf{KL}_2(\mathbb{Q})}{\lambda} + n\frac{\lambda}{2}\bar{V}_n(\mathbb{Q}),$$

with high probability. Recognizing that $\mathbb{E}_{\theta \sim \mathbb{Q}}[M_n(\theta)] = nI_3$, we derive the desired inequality

$$|I_3| \leq \frac{\mathsf{KL}_2(\mathbb{Q})}{n\lambda} + \frac{\lambda}{2}\bar{V}_n(\mathbb{Q}).$$

Our final bound is obtained by combining the previous inequalities on $|I_1|$, $|I_2|$, and $|I_3|$.

# 5 EXPERIMENTS

We present our core experiments in this section. Details and additional experiments are provided in Appendix C, where we also discuss the tightness of our bound in Appendix C.5. Our code is publicly available on GitHub.

## 5.1 SETTING

We adopt a similar setting to Sakhi et al. [2023]. We begin with a supervised training set $\mathcal{S}^{\mathrm{TR}}$ and convert it into logged bandit data $S$ using the standard supervised-to-bandit conversion method [Agarwal et al., 2014]. In this conversion, the label set $\mathcal{A}$ serves as the action space, while the input space serves as the context space $\mathcal{X}$. We then use $S$ to train our policies. After training, we evaluate the reward of the learned policies on the supervised test set $\mathcal{S}^{\mathrm{TS}}$. The resulting reward measures the ability of the learned policy to predict the true labels of the inputs in the test set and serves as our performance metric. We use two image classification datasets for this purpose: MNIST [LeCun et al., 1998] and FashionMNIST [Xiao et al., 2017]. Although we also explored the EMNIST dataset, it led to similar conclusions, so we did not include it to reduce clutter.

We define the logging policy as $\pi_0 = \pi_{\eta_0 \cdot \mu_0}^{\mathrm{SOF}}$ as in (10),

$$\pi_{\eta_0 \cdot \mu_0}^{\mathrm{SOF}}(a|x) = \frac{\exp(\eta_0 \phi(x)^\top \mu_{0,a})}{\sum_{a' \in \mathcal{A}} \exp(\eta_0 \phi(x)^\top \mu_{0,a'})}, \quad (16)$$

where $\mu_0 = (\mu_{0,a})_{a \in \mathcal{A}} \in \mathbb{R}^{dK}$ are learned using 5% of the training set $\mathcal{S}^{\mathrm{TR}}$. The parameter $\eta_0 \in \mathbb{R}$ is an inverse-temperature parameter that controls the quality of the logging policy $\pi_0$. Higher values of $\eta_0$ lead to a better-performing logging policy, while lower values lead to a poorer-performing logging policy. In particular, $\eta_0 = 0$ corresponds to a uniform logging policy. We set the prior as $\mathbb{P} = \mathcal{N}(\eta_0\mu_0, I_{dK})$ in all PAC-Bayesian learning principles considered in these experiments, including ours. We train policies on the remaining 95% of $\mathcal{S}^{\mathrm{TR}}$ using Adam [Kingma and Ba, 2014] with a learning rate of 0.1 for 20 epochs. The training objective for learning the policy varies based on the chosen method: we use our theoretical bound in (14), our proposed heuristic in (15), or other pessimism learning principles found in the literature.

We consider two main experiments. In Section 5.2, we focus on a common IW regularization technique, specifically Clip in (4). We then apply PAC-Bayesian learning principles from the literature that were specifically designed for Clip and compare them with ours applied to Clip. The goal is to demonstrate that our learning principle not only has broader applicability but also outperforms existing ones. After validating the improved performance of our PAC-Bayesian learning principle, we proceed in Section 5.3 to compare existing IW regularizations by training policies using our learning principles applied to them. The goal of these experiments is to determine whether there is a particular IW regularization technique that yields improved performance in OPL.

## 5.2 COMPARING LEARNING PRINCIPLES UNDER A COMMON IW REGULARIZATION

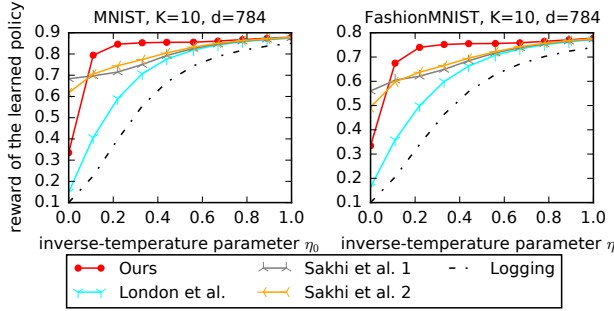

Figure 1: Performance of the learned policy with different PAC-Bayes pessimistic learning principles (our Corollary 2 and those in London and Sandler [2019], Sakhi et al. [2023]) using the Clip IPS risk estimator in (4) .

Here, we focus on the impact of different pessimistic learning principles on the performance of the learned policy given a fixed IW regularization method, specifically Clip as defined in (4). Recall that Clip regularizes the IW as $\hat{w}(x,a) = \frac{\pi(a|x)}{\max(\pi_0(a|x),\tau)}$, with $\tau$ set to $1/\sqrt[4]{n}$ following the suggestion in Ionides [2008]. To ensure a fair comparison, we consider PAC-Bayesian learning principles from the literature where the theoretical bound was optimized. Specifically, we include two PAC-Bayesian bounds proposed prior to our work for Clip, from London and Sandler [2019] and Sakhi et al. [2023]. We label the baselines as *London et al.* for optimizing the bound from London and Sandler [2019, Theorem 1], and for Sakhi et al. [2023], we distinguish their two bounds as *Sakhi et al. 1* (from Sakhi et al. [2023, Proposition 1], based on Catoni [2007]) and *Sakhi et al. 2* (from Sakhi et al. [2023, Proposition 3], a Bernstein-type bound). Since both London and Sandler [2019] and Sakhi et al. [2023] used the linear IW regularization trick described in Section 4, we compare their methods with optimizing our bound in (2), a direct consequence of Theorem 1 when the IW regularization is linear in $\pi$. Since we use linear IW regularizations, we optimize over Gaussian policies as described in Aouali et al. [2023a] and briefly discussed in Appendix C.3, as these are known to perform better in these scenarios [Sakhi et al., 2023, Aouali et al., 2023a]. Finally, we also include the logging policy as a baseline.

In Figure 1, the reward achieved by the learned policy is plotted as a function of the quality (i.e., performance) of the logging policy, $\eta_0 \in [0, 1]$. This comparison is conducted for learned policies that were optimized using one of the pessimistic learning principles above. The results demonstrate that ours outperforms all baselines across a wide range of logging policies. Thus, in addition to being generic and applicable to a large family of IW regularizers, our approach proves to be more effective than objectives tailored for specific IW regularizations. The enhanced performance of our method holds when $\eta_0$ is not very close to zero, a more realistic scenario in practical settings where the logging policy typically outperforms a uniform policy. Additionally, note that the performance of the learned policy using any method (including ours) improves upon the performance of the logging policy (indicated by dashed black lines).

Finally, we also conducted an experiment comparing our learning principle, **Heuristic Optimization**, with the $L_2$ heuristic from London and Sandler [2019]. We found that both heuristics had identical performance (Appendix C.4).

## 5.3 COMPARING IW REGULARIZATIONS UNDER A COMMON LEARNING PRINCIPLE

After demonstrating the favorable performance of our approach in the previous section, we now evaluate its performance with different IW regularization techniques. Specifically, we consider Clip, Har, IX, and ES as defined in (4). We employ both learning principles: one that directly optimizes the theoretical bound and another that optimizes the heuristic derived from it. For the bound optimization, we cannot use Corollary 2 since it includes a non-linear IW regularization (Har). Instead, we optimize the bound in Theorem 1 as explained in the **Bound Optimization** paragraph in Section 4.3. For the heuristic optimization, we use the method described in the **Heuristic Optimization** paragraph in Section 4.3. In this context, we optimize over softmax policies defined as

$$\pi_\theta^{\text{SOF}}(a|x) = \frac{\exp(\phi(x)^\top \theta_a)}{\sum_{a' \in \mathcal{A}} \exp(\phi(x)^\top \theta_{a'})}, \qquad (17)$$

where parameters $\theta$ are learned using either **Bound Optimization** (14) or **Heuristic Optimization** (15). **Bound Optimization** involves a single hyperparameter, $\lambda$, as defined in Theorem 1. We set $\lambda$ to its optimal value, $\lambda_*$, which minimizes the bound with respect to $\lambda$. Our theory does not support this approach since Theorem 1 requires $\lambda$ to be fixed in advance, whereas $\lambda_*$ is data-dependent. However, we found this method to yield good empirical performance. On the other hand, **Heuristic Optimization** relies on three hyperparameters, $\lambda_1$, $\lambda_2$, and $\lambda_3$, which we set to $\lambda_1 = \lambda_2 = \lambda_3 = 10^{-5}$.

In Figures 2 and 3, we present the rewards of the learned policies using different IW regularizations as a function of the quality of the logging policy $\pi_0$, based on the two proposed learning principles: **Bound Optimization** in Figure 2 and **Heuristic Optimization** in Figure 3. In both figures, the first and second rows correspond to results on MNIST and FashionMNIST, respectively. In the first four columns,

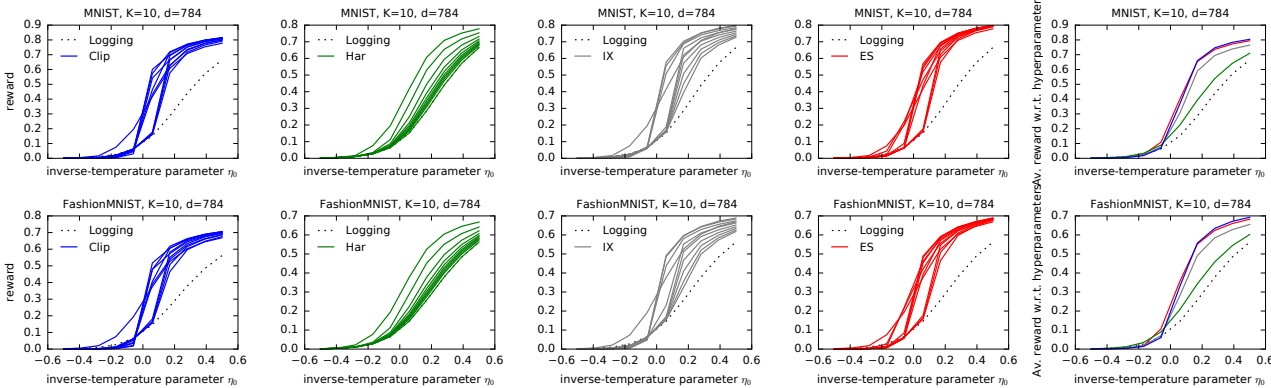

Figure 2: Performance of the policy learned by **Bound Optimization** (14) for different IW regularizations. The $x$-axis reflects the quality of the logging policy $\eta_0 \in [-0.5, 0.5]$. In the first four columns, we plot the reward of the learned policy using a fixed IW regularization technique (Clip, Har, IX, or ES as defined in (4)) for various values of its hyperparameter within $[0, 1]$. In the last column, we report the mean reward across these hyperparameter values.

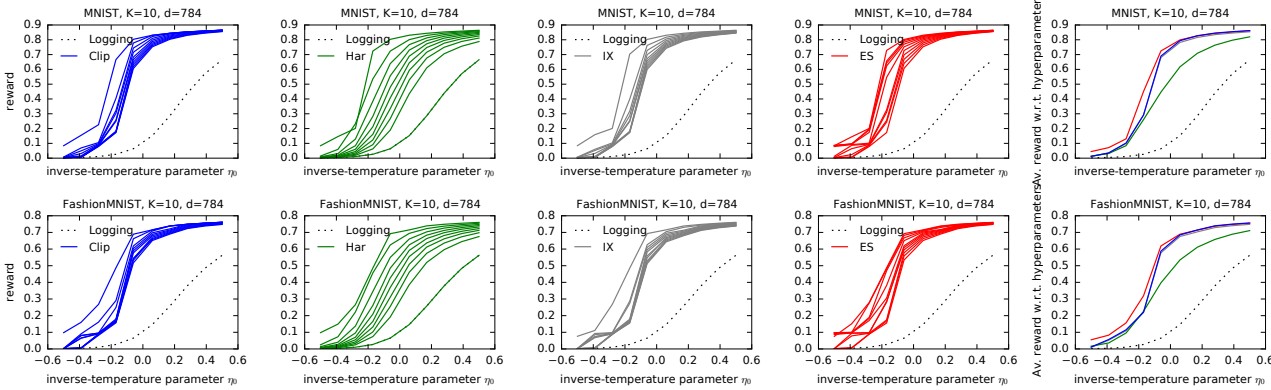

Figure 3: Performance of the policy learned by **Heuristic Optimization** (15) for different IW regularizations. The $x$-axis reflects the quality of the logging policy $\eta_0 \in [-0.5, 0.5]$. In the first four columns, we plot the reward of the learned policy using a fixed IW regularization technique (Clip, Har, IX, or ES as defined in (4)) for various values of its hyperparameter within $[0, 1]$. In the last column, we report the mean reward across these hyperparameter values.

we plot the reward of the learned policy using a fixed IW regularization technique (Clip, Har, IX, or ES as defined in (4)) for various values of its hyperparameter within $[0, 1]$. In the last column, we report the mean reward across these hyperparameter values to assess the sensitivity of the IW regularization technique to its hyperparameter. The $x$-axis represents $\eta_0$, which controls the quality of the logging policy; higher values indicate better performance of the logging policy. We vary $\eta_0 \in [-0.5, 0.5]$ to consider logging policies that perform worse than the uniform one (i.e., when $\eta_0 < 0$), to highlight settings that might require more IW regularization, although such scenarios may not be realistic.

Our results lead to the following conclusions. In Figure 2, we observe that all regularizations result in improved performance over the logging policy (i.e., all lines are above the dashed line representing the performance of the logging policy), with the Har regularization showing less im-

provement. Overall, Clip, IX, and ES achieve comparable performances, as summarized in the far-right column, despite regularizing IWs in very different ways. On the one hand, these results align with the generality of our bound, which applies to all these IW regularizations. On the other hand, they suggest that one can choose any IW regularization method when learning the policy by optimizing the theoretical bound without risking underperformance.

These results and conclusions are further confirmed by the rewards reported in Figure 3, where the policies are learned through **Heuristic Optimization** (15). The performances are even better than those obtained when optimizing the theoretical bound. As discussed in Section 4.3, this may be due to the practical optimization of the theoretical bound, where we used Monte Carlo to estimate the expectations, which performs poorly in high-dimensional problems. However, optimizing the heuristic or theoretical bound leads to

similar performance when the IW regularizers are linear (Appendix C.3) since, in that case, the Monte Carlo estimation was improved. Moreover, as summarized in the far-right column of Figure 3, the average performances for all regularizations are comparable, except for Har, which is below the others, and ES, which performs slightly better. Notably, for all regularizations, there is at least one choice of regularization hyperparameter that achieves optimal performance. This finding diverges from Aouali et al. [2023a], who attributed significant performance improvements to ES in a similar setting to ours. Our results clarify that these gains may be due to their newly introduced pessimistic learning principle rather than their smooth IW regularization (ES).

# 6 CONCLUSION

In this paper, we present the first comprehensive analysis of estimators that rely on IW regularization techniques, an increasingly popular approach in OPE and OPL. Our results hold for a broad spectrum of IW regularization methods and are also applicable to the standard IPS without any IW regularization. From our theoretical findings, we derive two learning principles that apply across various IW regularizations. Our results suggest that despite the numerous proposed IW regularization techniques for OPE, conventional methods like clipping, still perform very well in OPL.

Nevertheless, our work has three primary limitations. First, our bound includes empirical bias and variance terms, making it challenging to derive data-independent suboptimality gaps, as discussed in Section 4.2. Additionally, two-sided bounds for regularized IPS can be loose because they treat both tails similarly, whereas some studies indicate differences between the lower and upper tails of such estimators. Investigating methods to prove generic bounds by treating each side of the inequality individually, as in Gabbianelli et al. [2024], could address this issue and is left for future research. Second, optimizing our bound relies on the reparametrization trick and Monte Carlo estimation, which may have limitations in high-dimensional problems. Also, our reparametrization trick was only applied to simple linear-softmax policies defined in (10). Therefore, exploring more advanced techniques for optimizing our theoretical bound presents an intriguing direction for future research. While this limitation can be mitigated by considering linear IW regularization techniques, as discussed in Corollary 2, there is potential for better practical optimization of the bound for non-linear IW regularizations. Finally, extending our experiments to more complex policies and challenging settings, such as recommender systems with large action spaces, could further highlight the impact of IW regularization.

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

# Unified PAC-Bayesian Study of Pessimism for Offline Policy Learning with Regularized Importance Sampling
# (Supplementary Material)

**Imad Aouali**[1,2]  **Victor-Emmanuel Brunel**[1]  **David Rohde**[2]  **Anna Korba**[1]

[1]CREST, ENSAE, IP Paris, France
[2]Criteo AI Lab, Paris, France

The supplementary material is organized as follows.

- **Appendix A** includes an extended related work discussion.
- **Appendix B** outlines the proofs of our main results, as well as additional discussions.
- **Appendix C** presents our experimental setup for reproducibility, along with supplementary experiments.

## A  EXTENDED RELATED WORK

The framework of contextual bandits is a widely adopted model for addressing online learning in uncertain environments [Lattimore and Szepesvari, 2019, Auer et al., 2002, Thompson, 1933, Russo et al., 2018, Li et al., 2010, Chu et al., 2011]. This framework naturally aligns with the online learning paradigm, which seeks to adapt in real-time. However, in practical scenarios, challenges arise when dealing with a large action space. While numerous online algorithms have emerged to efficiently navigate the large action spaces in contextual bandits [Zong et al., 2016, Hong et al., 2022, Zhu et al., 2022, Aouali et al., 2023b], a notable need remains for offline methods that enable the optimization of decision-making based on historical data. Fortunately, we often possess large sample sets summarizing historical interactions with contextual bandit environments. Leveraging this, agents can enhance their policies offline [Swaminathan and Joachims, 2015a, London and Sandler, 2019, Sakhi et al., 2023, Aouali et al., 2023a]. This study is primarily dedicated to exploring this offline mode of contextual bandits, often referred to as the *off-policy* formulation [Dudík et al., 2011, 2012, Dudik et al., 2014, Wang et al., 2017, Farajtabar et al., 2018]. Off-policy contextual bandits entail two primary tasks. The first task, known as *off-policy evaluation (OPE)*, revolves around estimating policy performance using historical data. This estimation replicates how evaluations would unfold as if the policy is engaging with the environment in real-time. Subsequently, the derived estimator is optimized to find the optimal policy, and this is called *off-policy learning (OPL)* [Swaminathan and Joachims, 2015a]. Next, we review both OPE and OPL.

### A.1  OFF-POLICY EVALUATION

In recent years, OPE has experienced a noticeable surge of interest, with numerous significant contributions [Dudík et al., 2011, 2012, Dudik et al., 2014, Wang et al., 2017, Farajtabar et al., 2018, Su et al., 2019, 2020, Kallus et al., 2021, Metelli et al., 2021, Kuzborskij et al., 2021, Saito and Joachims, 2022, Sakhi et al., 2020, Jeunen and Goethals, 2021, Saito et al., 2023]. The literature on OPE can be broadly classified into three primary approaches. The first, referred to as the direct method (DM) [Jeunen and Goethals, 2021, Aouali et al., 2024], involves the development of a model designed to approximate expected costs for any context-action pair. This model is subsequently employed to estimate the performance of the policies. This approach is often used in large-scale recommender systems [Sakhi et al., 2020, Jeunen and Goethals, 2021, Aouali et al., 2022b,a]. The second approach, known as inverse propensity scoring (IPS) [Horvitz and Thompson, 1952, Dudík et al., 2012], aims to estimate the costs associated with the evaluated policies by correcting for the inherent preference bias of the logging policy within the sample dataset. While IPS maintains its unbiased nature when operating under the assumption that the evaluation policy is absolutely continuous concerning the logging policy, it can be susceptible to high variance

and substantial bias when this assumption is violated [Sachdeva et al., 2020]. In response to the variance issue, various techniques have been introduced, including the clipping of importance weights [Ionides, 2008, Swaminathan and Joachims, 2015a], their smoothing [Aouali et al., 2023a], and self-normalization [Swaminathan and Joachims, 2015b], among others [Gilotte et al., 2018]. The third approach, known as doubly robust (DR) [Robins and Rotnitzky, 1995, Bang and Robins, 2005, Dudík et al., 2011, Dudik et al., 2014, Farajtabar et al., 2018], combines elements from both the direct method (DM) and inverse propensity scoring (IPS). This amalgamation serves to reduce variance in the estimation process. Typically, the accuracy of an OPE estimator $\hat{R}(\pi, S)$, is assessed using the mean squared error (MSE). It's worth mentioning that Metelli et al. [2021] advocate for the preference of high-probability concentration rates as the favored metric for evaluating OPE estimators. This work focuses primarily on OPL and hence we did not evaluate the regularized IPS on OPE.

## A.2 OFF-POLICY LEARNING

Prior OPL research has primarily focused on the derivation of learning principles rooted in generalization bounds *under the clipped IPS* estimator. First, Swaminathan and Joachims [2015a] designed a learning principle that favors policies that simultaneously demonstrate low estimated cost and empirical variance. Furthermore, Faury et al. [2020] extended this concept by incorporating distributional robustness optimization, while Zenati et al. [2020] adapted it to continuous action spaces. The latter also proposed a softer importance weight regularization that is different from clipping. Additionally, London and Sandler [2019] has elegantly established a connection between PAC-Bayes theory and OPL. This connection led to the derivation of a novel PAC-Bayes generalization bound for the clipped IPS. Once again, this bound served as the foundation for the creation of a novel learning principle that promotes policies with low estimated cost and parameters that are in close proximity to those of the logging policy in terms of $L_2$ distance. Additionally, Sakhi et al. [2023] introduced new generalization bounds tailored to the clipped IPS. A notable feature of their approach is the direct optimization of the theoretical bound, rendering the use of learning principles unnecessary. Expanding upon these advancements, Aouali et al. [2023a] presented a generalized bound designed for IPS with exponential smoothing. What distinguishes this particular bound from previous ones is its applicability to standard IPS without the prerequisite assumption that importance weights are bounded. They also demonstrated that optimizing this bound for IPS with exponential smoothing results in superior performance compared to optimizing existing bounds for clipped IPS. However, a significant question lingers: is the performance improvement primarily attributed to the enhanced regularization offered by exponential smoothing over clipping, or is it the consequence of the novel bound itself? This uncertainty arises from the fact that the bounds employed for clipping and exponential smoothing differ. In light of these considerations, our work introduces a unified set of generalization bounds, allowing for meaningful comparisons that address this question and contribute to a deeper understanding of the performance dynamics.

## B MISSING PROOFS AND RESULTS

In this section, we prove Theorem 1.

**Theorem 3** (Theorem 1 Restated). *Let $\lambda > 0$, $n \geq 1$, $\delta \in (0, 1)$, and let $\mathbb{P}$ be a fixed prior on $\Theta$. Then the following inequality holds with probability at least $1 - \delta$ for any distribution $\mathbb{Q}$ on $\Theta$*

$$\left| \mathbb{E}_{\theta \sim \mathbb{Q}} \left[ R(\pi_\theta) - \hat{R}(\pi_\theta, S) \right] \right| \leq \sqrt{\frac{\mathrm{KL}_1(\mathbb{Q})}{2n}} + B_n(\mathbb{Q}) + \frac{\mathrm{KL}_2(\mathbb{Q})}{n\lambda} + \frac{\lambda}{2} \bar{V}_n(\mathbb{Q}), \tag{18}$$

*where $\mathrm{KL}_1(\mathbb{Q}) = D_{\mathrm{KL}}(\mathbb{Q} \| \mathbb{P}) + \log \frac{4\sqrt{n}}{\delta}$, $\mathrm{KL}_2(\mathbb{Q}) = D_{\mathrm{KL}}(\mathbb{Q} \| \mathbb{P}) + \log(4/\delta)$, and*

$$\bar{V}_n(\mathbb{Q}) = \frac{1}{n} \sum_{i=1}^{n} \mathbb{E}_{\theta \sim \mathbb{Q}} [\mathbb{E}_{a \sim \pi_0(\cdot | x_i)} \left[ \hat{w}_\theta(x_i, a)^2 \right] + \hat{w}_\theta(x_i, a_i)^2 c_i^2]$$

$$B_n(\mathbb{Q}) = \frac{1}{n} \sum_{i=1}^{n} \sum_{a \in \mathcal{A}} \mathbb{E}_{\theta \sim \mathbb{Q}} [|\pi_\theta(a | x_i) - \pi_0(a | x_i) \hat{w}_\theta(x_i, a)|]$$

*Proof.* First, we decompose the difference $\mathbb{E}_{\theta\sim\mathbb{Q}}\left[R(\pi_\theta) - \hat{R}(\pi_\theta, S)\right]$ as

$$\mathbb{E}_{\theta\sim\mathbb{Q}}\left[R(\pi_\theta) - \hat{R}(\pi_\theta, S)\right] = \underbrace{\mathbb{E}_{\theta\sim\mathbb{Q}}\left[R(\pi_\theta) - \frac{1}{n}\sum_{i=1}^n R(\pi_\theta|x_i)\right]}_{I_1} + \underbrace{\frac{1}{n}\sum_{i=1}^n \mathbb{E}_{\theta\sim\mathbb{Q}}\left[R(\pi_\theta|x_i) - \frac{1}{n}\sum_{i=1}^n \hat{R}(\pi_\theta|x_i)\right]}_{I_2}$$

$$+ \underbrace{\frac{1}{n}\sum_{i=1}^n \mathbb{E}_{\theta\sim\mathbb{Q}}\left[\hat{R}(\pi_\theta|x_i)\right] - \mathbb{E}_{\theta\sim\mathbb{Q}}\left[\hat{R}(\pi_\theta, S)\right]}_{I_3},$$

where

$$R(\pi_\theta) = \mathbb{E}_{x\sim\nu, a\sim\pi_\theta(\cdot|x)}\left[c(x,a)\right],$$
$$R(\pi_\theta|x_i) = \mathbb{E}_{a\sim\pi_\theta(\cdot|x_i)}\left[c(x_i,a)\right],$$
$$\hat{R}(\pi_\theta|x_i) = \mathbb{E}_{a\sim\pi_0(\cdot|x_i)}\left[\hat{w}_\theta(x_i,a)c(x_i,a)\right],$$
$$\hat{R}(\pi_\theta, S) = \frac{1}{n}\sum_{i=1}^n \hat{w}_\theta(x_i,a_i)c_i,$$

where $\hat{w}_\theta(x,a) = g(\pi_\theta(a|x), \pi_0(a|x))$ for some non-negative function $g$. Our goal is to bound $|\mathbb{E}_{\theta\sim\mathbb{Q}}\left[R(\pi_\theta) - \hat{R}(\pi_\theta, S)\right]|$ and thus we need to bound $|I_1| + |I_2| + |I_3|$. We start with $|I_1|$, Alquier [2021, Theorem 3.3] yields that following inequality holds with probability at least $1 - \delta/2$ for any distribution $\mathbb{Q}$ on $\Theta$

$$|I_1| \leq \sqrt{\frac{D_{\mathrm{KL}}(\mathbb{Q}\|\mathbb{P}) + \log\frac{4\sqrt{n}}{\delta}}{2n}},$$
$$= \sqrt{\frac{\mathrm{KL}_1(\mathbb{Q})}{2n}}. \tag{19}$$

Moreover, $|I_2|$ can be bounded by decomposing it as

$$|I_2| = \left|\mathbb{E}_{\theta\sim\mathbb{Q}}\left[\frac{1}{n}\sum_{i=1}^n \mathbb{E}_{a\sim\pi_\theta(\cdot|x_i)}\left[c(x_i,a)\right] - \frac{1}{n}\sum_{i=1}^n \mathbb{E}_{a\sim\pi_0(\cdot|x_i)}\left[\hat{w}_\theta(x_i,a)c(x_i,a)\right]\right]\right|,$$
$$= \left|\frac{1}{n}\sum_{i=1}^n \sum_{a\in\mathcal{A}} \mathbb{E}_{\theta\sim\mathbb{Q}}\left[\pi_\theta(a|x_i)c(x_i,a) - \pi_0(a|x_i)\hat{w}_\theta(x_i,a)c(x_i,a)\right]\right|,$$
$$\leq \frac{1}{n}\sum_{i=1}^n \sum_{a\in\mathcal{A}} \mathbb{E}_{\theta\sim\mathbb{Q}}\left[\left|\pi_\theta(a|x_i) - \pi_0(a|x_i)\hat{w}_\theta(x_i,a)\right||c(x_i,a)|\right].$$

But $|c(x,a)| \leq 1$ for any $a \in \mathcal{A}$ and $x \in \mathcal{X}$. Thus

$$|I_2| \leq \frac{1}{n}\sum_{i=1}^n \sum_{a\in\mathcal{A}} \mathbb{E}_{\theta\sim\mathbb{Q}}\left[\left|\pi_\theta(a|x_i) - \pi_0(a|x_i)\hat{w}_\theta(x_i,a)\right|\right],$$
$$= B_n(\mathbb{Q}) \tag{20}$$

Finally, we need to bound the main term $|I_3|$. To achieve this, we borrow and adapt the statement of the following technical lemma [Haddouche and Guedj, 2022, Theorem 2.1] to our setting.

**Lemma 4.** *Let $\mathcal{Z}$ be an instance space and let $S_n = (z_i)_{i\in[n]}$ be an n-sized dataset for some $n \geq 1$. Let $(\mathcal{F}_i)_{i\in\{0\}\cup[n]}$ be a filtration adapted to $S_n$. Also, let $\Theta$ be a parameter space and $\pi_\theta$ for $\theta \in \Theta$ are the corresponding policies. Now assume that $(f_i(S_i, \pi_\theta))_{i\in[n]}$ is a martingale difference sequence for any $\theta \in \Theta$, that is for any $i \in [n]$, and $\theta \in \Theta$, we have that $\mathbb{E}\left[f_i(S_i, \pi_\theta)|\mathcal{F}_{i-1}\right] = 0$. Moreover, for any $\theta \in \Theta$, let $M_n(\theta) = \sum_{i=1}^n f_i(S_i, \pi_\theta)$. Then for any fixed prior, $\mathbb{P}$, on $\Theta$, any $\lambda > 0$, the following holds with probability $1 - \delta$ over the sample $S_n$, simultaneously for any $\mathbb{Q}$, on $\Theta$*

$$|\mathbb{E}_{\theta\sim\mathbb{Q}}\left[M_n(\theta)\right]| \leq \frac{D_{\mathrm{KL}}(\mathbb{Q}\|\mathbb{P}) + \log(2/\delta)}{\lambda} + \frac{\lambda}{2}\left(\mathbb{E}_{\theta\sim\mathbb{Q}}\left[\langle M\rangle_n(\theta) + [M]_n(\theta)\right]\right),$$

*where $\langle M\rangle_n(\theta) = \sum_{i=1}^n \mathbb{E}\left[f_i(S_i, \pi_\theta)^2|\mathcal{F}_{i-1}\right]$ and $[M]_n(\theta) = \sum_{i=1}^n f_i(S_i, \pi_\theta)^2$.*

Recall that $\hat{w}_\theta(x, a) = g(\pi_\theta(a|x), \pi_0(a|x))$ for some non-negative function $g$. To apply Lemma 4, we need to construct an adequate martingale difference sequence $(f_i(S_i, \pi_\theta))_{i \in [n]}$ for $\theta \in \Theta$ that allows us to retrieve $|I_3|$. To achieve this, we define $S_n = (a_i)_{i \in [n]}$ as the set of $n$ taken actions. Also, we let $(\mathcal{F}_i)_{i \in \{0\} \cup [n]}$ be a filtration adapted to $S_n$. For $\theta \in \Theta$, we define $f_i(S_i, \pi_\theta)$ as

$$f_i(S_i, \pi_\theta) = f_i(a_i, \pi_\theta) = \mathbb{E}_{a \sim \pi_0(\cdot|x_i)}\left[g(\pi_\theta(a|x_i), \pi_0(a|x_i))c(x_i, a)\right] - g(\pi_\theta(a_i|x_i), \pi_0(a_i|x_i))c(x_i, a_i).$$

We stress that $f_i(S_i, \pi_\theta)$ only depends on the last action in $S_i$, $a_i$, and the policy $\pi_\theta$. For this reason, we denote it by $f_i(a_i, \pi_\theta)$. The function $f_i$ is indexed by $i$ since it depends on the fixed $i$-th context, $x_i$. The context $x_i$ is fixed and thus randomness only comes from $a_i \sim \pi_0(\cdot|x_i)$. It follows that the expectations are under $a_i \sim \pi_0(\cdot|x_i)$. First, we have that $\mathbb{E}\left[f_i(a_i, \pi_\theta)|\mathcal{F}_{i-1}\right] = 0$ for any $i \in [n]$, and $\theta \in \Theta$. This follows from

$$\mathbb{E}\left[f_i(a_i, \pi_\theta)|\mathcal{F}_{i-1}\right] = \mathbb{E}_{a_i \sim \pi_0(\cdot|x_i)}\left[f_i(a_i, \pi_\theta)\Big|a_1, \ldots, a_{i-1}\right],$$

$$= \mathbb{E}_{a_i \sim \pi_0(\cdot|x_i)}\left[\mathbb{E}_{a \sim \pi_0(\cdot|x_i)}\left[g(\pi_\theta(a|x_i), \pi_0(a|x_i))c(x_i, a)\right] - g(\pi_\theta(a_i|x_i), \pi_0(a_i|x_i))c(x_i, a_i)\Big|a_1, \ldots, a_{i-1}\right],$$

$$\overset{(i)}{=} \mathbb{E}_{a \sim \pi_0(\cdot|x_i)}\left[g(\pi_\theta(a|x_i), \pi_0(a|x_i))c(x_i, a)\right] - \mathbb{E}_{a_i \sim \pi_0(\cdot|x_i)}\left[g(\pi_\theta(a_i|x_i), \pi_0(a_i|x_i))c(x_i, a_i)\Big|a_1, \ldots, a_{i-1}\right].$$

In $(i)$ we use the fact that given $x_i$, $\mathbb{E}_{a \sim \pi_0(\cdot|x_i)}\left[g(\pi_\theta(a|x_i), \pi_0(a|x_i))c(x_i, a)\right]$ is deterministic. Now $a_i$ does not depend on $a_1, \ldots, a_{i-1}$ since logged data is i.d.d. Hence

$$\mathbb{E}_{a_i \sim \pi_0(\cdot|x_i)}\left[g(\pi_\theta(a_i|x_i), \pi_0(a_i|x_i))c(x_i, a_i)\Big|a_1, \ldots, a_{i-1}\right] = \mathbb{E}_{a_i \sim \pi_0(\cdot|x_i)}\left[g(\pi_\theta(a_i|x_i), \pi_0(a_i|x_i))c(x_i, a_i)\right],$$

$$= \mathbb{E}_{a \sim \pi_0(\cdot|x_i)}\left[g(\pi_\theta(a|x_i), \pi_0(a|x_i))c(x_i, a)\right].$$

It follows that

$$\mathbb{E}[f_i(a_i, \pi_\theta)|\mathcal{F}_{i-1}]$$
$$= \mathbb{E}_{a \sim \pi_0(\cdot|x_i)}\left[g(\pi_\theta(a|x_i), \pi_0(a|x_i))c(x_i, a)\right] - \mathbb{E}_{a_i \sim \pi_0(\cdot|x_i)}\left[g(\pi_\theta(a_i|x_i), \pi_0(a_i|x_i))c(x_i, a_i)\Big|a_1, \ldots, a_{i-1}\right],$$
$$= \mathbb{E}_{a \sim \pi_0(\cdot|x_i)}\left[g(\pi_\theta(a|x_i), \pi_0(a|x_i))c(x_i, a)\right] - \mathbb{E}_{a \sim \pi_0(\cdot|x_i)}\left[g(\pi_\theta(a|x_i), \pi_0(a|x_i))c(x_i, a)\right],$$
$$= 0.$$

Therefore, for any $\theta \in \Theta$, $(f_i(a_i, \pi_\theta))_{i \in [n]}$ is a martingale difference sequence. Hence we apply Lemma 4 and obtain that the following inequality holds with probability at least $1 - \delta/2$ for any $\mathbb{Q}$ on $\Theta$

$$|\mathbb{E}_{\theta \sim \mathbb{Q}}\left[M_n(\theta)\right]| \leq \frac{D_{\mathrm{KL}}(\mathbb{Q}\|\mathbb{P}) + \log(4/\delta)}{\lambda} + \frac{\lambda}{2}\left(\mathbb{E}_{\theta \sim \mathbb{Q}}\left[\langle M\rangle_n(\theta) + [M]_n(\theta)\right]\right),$$

$$= \frac{\mathrm{KL}2(\mathbb{Q})}{\lambda} + \frac{\lambda}{2}\left(\mathbb{E}_{\theta \sim \mathbb{Q}}\left[\langle M\rangle_n(\theta) + [M]_n(\theta)\right]\right), \tag{21}$$

where

$$M_n(\theta) = \sum_{i=1}^n f_i(a_i, \pi_\theta),$$

$$\langle M\rangle_n(\theta) = \sum_{i=1}^n \mathbb{E}\left[f_i(a_i, \pi_\theta)^2|\mathcal{F}_{i-1}\right],$$

$$[M]_n(\theta) = \sum_{i=1}^n f_i(a_i, \pi_\theta)^2.$$

Now these terms can be decomposed as

$$\mathbb{E}_{\theta \sim \mathbb{Q}}\left[M_n(\theta)\right] = \sum_{i=1}^{n} \mathbb{E}_{\theta \sim \mathbb{Q}}\left[f_i\left(a_i, \pi_\theta\right)\right],$$

$$= \sum_{i=1}^{n} \mathbb{E}_{\theta \sim \mathbb{Q}}\left[\mathbb{E}_{a \sim \pi_0(\cdot|x_i)}\left[g(\pi_\theta(a|x_i), \pi_0(a|x_i))c(x_i, a)\right] - g(\pi_\theta(a_i|x_i), \pi_0(a_i|x_i))c(x_i, a_i)\right],$$

$$\stackrel{(i)}{=} \sum_{i=1}^{n} \mathbb{E}_{\theta \sim \mathbb{Q}}\left[\hat{R}(\pi_\theta|x_i)\right] - n\mathbb{E}_{\theta \sim \mathbb{Q}}\left[\hat{R}(\pi_\theta, S)\right],$$

$$= nI_3, \tag{22}$$

where we used the fact that $c_i = c(a_i, x_i)$ for any $i \in [n]$ in $(i)$.

Now we focus on the terms $\langle M \rangle_n(\theta)$ and $[M]_n(\theta)$. First, we have that

$$f_i\left(a_i, \pi_\theta\right)^2 = \left(\mathbb{E}_{a \sim \pi_0(\cdot|x_i)}\left[g(\pi_\theta(a|x_i), \pi_0(a|x_i))c(x_i, a)\right] - g(\pi_\theta(a_i|x_i), \pi_0(a_i|x_i))c(x_i, a_i)\right)^2, \tag{23}$$

$$= \mathbb{E}_{a \sim \pi_0(\cdot|x_i)}\left[g(\pi_\theta(a|x_i), \pi_0(a|x_i))c(x_i, a)\right]^2 + \left(g(\pi_\theta(a_i|x_i), \pi_0(a_i|x_i))c(x_i, a_i)\right)^2$$
$$- 2\mathbb{E}_{a \sim \pi_0(\cdot|x_i)}\left[g(\pi_\theta(a|x_i), \pi_0(a|x_i))c(x_i, a)\right]g(\pi_\theta(a_i|x_i), \pi_0(a_i|x_i))c(x_i, a_i),$$

$$= \mathbb{E}_{a \sim \pi_0(\cdot|x_i)}\left[g(\pi_\theta(a|x_i), \pi_0(a|x_i))c(x_i, a)\right]^2 + g(\pi_\theta(a_i|x_i), \pi_0(a_i|x_i))^2 c(x_i, a_i)^2$$
$$- 2\mathbb{E}_{a \sim \pi_0(\cdot|x_i)}\left[g(\pi_\theta(a|x_i), \pi_0(a|x_i))c(x_i, a)\right]g(\pi_\theta(a_i|x_i), \pi_0(a_i|x_i))c(x_i, a_i).$$

Moreover, $f_i\left(a_i, \pi_\theta\right)^2$ does not depend on $a_1, \ldots, a_{i-1}$. Thus

$$\mathbb{E}\left[f_i\left(a_i, \pi_\theta\right)^2|\mathcal{F}_{i-1}\right] = \mathbb{E}_{a_i \sim \pi_0(\cdot|x_i)}\left[f_i\left(a_i, \pi_\theta\right)^2|\mathcal{F}_{i-1}\right] = \mathbb{E}_{a_i \sim \pi_0(\cdot|x_i)}\left[f_i\left(a_i, \pi_\theta\right)^2\right] = \mathbb{E}_{a \sim \pi_0(\cdot|x_i)}\left[f_i\left(a, h\right)^2\right].$$

Computing $\mathbb{E}_{a \sim \pi_0(\cdot|x_i)}\left[f_i\left(a, h\right)^2\right]$ using the decomposition in (23) yields

$$\mathbb{E}[f_i\left(a_i, \pi_\theta\right)^2|\mathcal{F}_{i-1}] = \mathbb{E}_{a \sim \pi_0(\cdot|x_i)}\left[f_i\left(a, h\right)^2\right],$$

$$= -\mathbb{E}_{a \sim \pi_0(\cdot|x_i)}\left[g(\pi_\theta(a|x_i), \pi_0(a|x_i))c(x_i, a)\right]^2 + \mathbb{E}_{a \sim \pi_0(\cdot|x_i)}\left[g(\pi_\theta(a|x_i), \pi_0(a|x_i))^2 c(x_i, a)^2\right] \tag{24}$$

Combining (23) and (24) leads to

$$\mathbb{E}[f_i\left(a_i, \pi_\theta\right)^2|\mathcal{F}_{i-1}] + f_i\left(a_i, \pi_\theta\right)^2 = \mathbb{E}_{a \sim \pi_0(\cdot|x_i)}\left[g(\pi_\theta(a|x_i), \pi_0(a|x_i))^2 c(x_i, a)^2\right] + g(\pi_\theta(a_i|x_i), \pi_0(a_i|x_i))^2 c(x_i, a_i)^2$$
$$- 2\mathbb{E}_{a \sim \pi_0(\cdot|x_i)}\left[g(\pi_\theta(a|x_i), \pi_0(a|x_i))c(x_i, a)\right]g(\pi_\theta(a_i|x_i), \pi_0(a_i|x_i))c(x_i, a_i),$$

$$\stackrel{(i)}{\leq} \mathbb{E}_{a \sim \pi_0(\cdot|x_i)}\left[g(\pi_\theta(a|x_i), \pi_0(a|x_i))^2 c(x_i, a)^2\right] + g(\pi_\theta(a_i|x_i), \pi_0(a_i|x_i))^2 c(x_i, a_i)^2. \tag{25}$$

The inequality in $(i)$ holds because $-2\mathbb{E}_{a \sim \pi_0(\cdot|x_i)}\left[g(\pi_\theta(a|x_i), \pi_0(a|x_i))c(x_i, a)\right]g(\pi_\theta(a_i|x_i), \pi_0(a_i|x_i))c(x_i, a_i) \leq 0$ since $g$ is non-negative. Therefore, we have that

$$\langle M \rangle_n(\theta) + [M]_n(\theta) \leq \sum_{i=1}^{n} \mathbb{E}_{a \sim \pi_0(\cdot|x_i)}\left[g(\pi_\theta(a|x_i), \pi_0(a|x_i))^2 c(x_i, a)^2\right] + g(\pi_\theta(a_i|x_i), \pi_0(a_i|x_i))^2 c(x_i, a_i)^2.$$

It follows that

$$\mathbb{E}_{\theta \sim \mathbb{Q}}\left[\langle M \rangle_n(\theta) + [M]_n(\theta)\right]$$

$$\leq \sum_{i=1}^{n} \mathbb{E}_{\theta \sim \mathbb{Q}}\left[\mathbb{E}_{a \sim \pi_0(\cdot|x_i)}\left[g(\pi_\theta(a|x_i), \pi_0(a|x_i))^2 c(x_i, a)^2\right]\right] + \mathbb{E}_{\theta \sim \mathbb{Q}}\left[g(\pi_\theta(a_i|x_i), \pi_0(a_i|x_i))^2 c(x_i, a_i)^2\right]. \tag{26}$$

Combining (21) and (26) yields

$$n|I_3| = |\sum_{i=1}^{n} \mathbb{E}_{\theta \sim \mathbb{Q}} \left[ \hat{R}(\pi_\theta | x_i) \right] - n\mathbb{E}_{\theta \sim \mathbb{Q}} \left[ \hat{R}(\pi_\theta, S) \right] |$$

$$\leq \frac{\text{KL}_2(\mathbb{Q})}{\lambda} + \frac{\lambda}{2} \sum_{i=1}^{n} \mathbb{E}_{\theta \sim \mathbb{Q}} \left[ \mathbb{E}_{a \sim \pi_0(\cdot | x_i)} \left[ g(\pi_\theta(a|x_i), \pi_0(a|x_i))^2 c(x_i, a)^2 \right] \right]$$

$$+ \mathbb{E}_{\theta \sim \mathbb{Q}} \left[ g(\pi_\theta(a_i|x_i), \pi_0(a_i|x_i))^2 c(x_i, a_i)^2 \right] . \qquad (27)$$

This means that the following inequality holds with probability at least $1 - \delta/2$ for any distribution $\mathbb{Q}$ on $\Theta$

$$|I_3| \leq \frac{\text{KL}_2(\mathbb{Q})}{n\lambda} + \frac{\lambda}{2n} \sum_{i=1}^{n} \mathbb{E}_{\theta \sim \mathbb{Q}} \left[ \mathbb{E}_{a \sim \pi_0(\cdot | x_i)} \left[ g(\pi_\theta(a|x_i), \pi_0(a|x_i))^2 c(x_i, a)^2 \right] \right]$$

$$+ \frac{\lambda}{2n} \sum_{i=1}^{n} \mathbb{E}_{\theta \sim \mathbb{Q}} \left[ g(\pi_\theta(a_i|x_i), \pi_0(a_i|x_i))^2 c(x_i, a_i)^2 \right] . \qquad (28)$$

However we know that $c(x, a)^2 \leq 1$ for any $x \in \mathcal{X}$ and $a \in \mathcal{A}$ and that $c(x_i, a_i) = c_i$ for any $i \in [n]$. Thus the following inequality holds with probability at least $1 - \delta/2$ for any distribution $\mathbb{Q}$ on $\Theta$

$$|I_3| \leq \frac{\text{KL}_2(\mathbb{Q})}{n\lambda} + \frac{\lambda}{2n} \sum_{i=1}^{n} \mathbb{E}_{\theta \sim \mathbb{Q}} \left[ \mathbb{E}_{a \sim \pi_0(\cdot | x_i)} \left[ g(\pi_\theta(a|x_i), \pi_0(a|x_i))^2 \right] \right] + \frac{\lambda}{2n} \sum_{i=1}^{n} \mathbb{E}_{\theta \sim \mathbb{Q}} \left[ g(\pi_\theta(a_i|x_i), \pi_0(a_i|x_i))^2 c_i^2 \right] ,$$

$$= \frac{\text{KL}_2(\mathbb{Q})}{n\lambda} + \frac{\lambda}{2} \bar{V}_n(\mathbb{Q}) , \qquad (29)$$

where we use that $g(\pi_\theta(a|x), \pi_0(a|x)) = \hat{w}_\theta(x, a)$. The union bound of (19) and (29) combined with the deterministic result in (20) yields that the following inequality holds with probability at least $1 - \delta$ for any distribution $\mathbb{Q}$ on $\Theta$

$$|\mathbb{E}_{\theta \sim \mathbb{Q}} \left[ R(\pi_\theta) - \hat{R}(\pi_\theta, S) \right] | \leq \sqrt{\frac{\text{KL}_1(\mathbb{Q})}{2n}} + B_n(\mathbb{Q}) + \frac{\text{KL}_2(\mathbb{Q})}{n\lambda} + \frac{\lambda}{2} \bar{V}_n(\mathbb{Q}) . \qquad (30)$$

$\square$

# C   ADDITIONAL EXPERIMENTS

## C.1   DETAILED SETUP

We begin by employing the well-established supervised-to-bandit conversion method as described in Agarwal et al. [2014]. Specifically, we work with two sets from a classification dataset: the training set denoted as $\mathcal{S}^{\text{TR}}$ and the testing set as $\mathcal{S}^{\text{TS}}$. The first step involves transforming the training set, $\mathcal{S}^{\text{TR}}$, into a bandit logged data denoted as $S$, following the procedure outlined in Algorithm 1. This newly created logged data, $S$, is subsequently employed to train our policies. The next phase assesses the effectiveness of the learned policies on the testing set, $\mathcal{S}^{\text{TS}}$, as outlined in Algorithm 2. We measure the performance of these policies using the reward obtained by running Algorithm 2. Higher rewards indicate superior performance. In our experimental evaluations, we make use of various image classification datasets, specifically: MNIST [LeCun et al., 1998] and FashionMNIST [Xiao et al., 2017].

The input to Algorithm 1 is a logging policy denoted as $\pi_0$, defined as follows

$$\pi_0(a|x) = \frac{\exp(\eta_0 \phi(x)^\top \mu_{0,a})}{\sum_{a' \in \mathcal{A}} \exp(\eta_0 \phi(x)^\top \mu_{0,a'})} , \qquad \forall (x, a) \in \mathcal{X} \times \mathcal{A} . \qquad (31)$$

Here, $\phi(x) \in \mathbb{R}^d$ represents the feature transformation function, which writes $\phi(x) = \frac{x}{\|x\|}$. The parameters $\mu_0 = (\mu_{0,a})_{a \in \mathcal{A}} \in \mathbb{R}^{dK}$ are learned using a fraction (5%) of the training set $\mathcal{S}^{\text{TR}}$, with the cross-entropy loss. Optimization is carried out using the Adam optimizer [Kingma and Ba, 2014]. The inverse-temperature parameter $\eta_0$ is a critical factor affecting the performance of the logging policy. A high positive value of $\eta_0$ indicates a well-performing logging policy, while a negative value leads to a lower-performing one. In all experiments, the prior $\mathbb{P}$ is set as $\mathbb{P} = \mathcal{N}(\eta_0 \mu_0, I_{dK})$.

**Algorithm 1** Supervised-to-bandit: creating the logged data

**Input.** Classification training dataset $\mathcal{S}^{\text{TR}} = \{(x_i, y_i)\}_{i=1}^{n}$, logging policy $\pi_0$.
**Output.** logged data $S = (x_i, a_i, c_i)_{i \in [n]}$.
Initialize $S = \{\}$
**for** $i = 1, \ldots, n$ **do**
$\quad a_i \sim \pi_0(\cdot | x_i)$
$\quad c_i = -\mathbb{I}_{\{a_i = y_i\}}$
$\quad S \leftarrow S \cup \{(x_i, a_i, c_i)\}$.

---

**Algorithm 2** Supervised-to-bandit: testing policies

**Input:** Classification test dataset $\mathcal{S}^{\text{TS}} = \{(x_i, y_i)\}_{i=1}^{n_{\text{TS}}}$, learned policy $\hat{\pi}_n$.
**Output:** Test reward $r$.
**for** $i = 1, \ldots, n_{\text{TS}}$ **do**
$\quad a_i \sim \hat{\pi}_n(\cdot | x_i)$
$\quad r_i = \mathbb{I}_{\{a_i = y_i\}}$
$r = \frac{1}{n_{\text{TS}}} \sum_{i=1}^{n_{\text{TS}}} r_i$.

---

## C.2 BOUND OPTIMIZATION

In this section, the learned policy $\hat{\pi}_n$ is obtained by optimizing the following objective

$$\operatorname*{argmax}_{\mathbb{Q}} \mathbb{E}_{\theta \sim \mathbb{Q}} \left[ \hat{R}(\pi_\theta, S) \right] + \sqrt{\frac{\text{KL}_1(\mathbb{Q})}{2n} + B_n(\mathbb{Q})} + \frac{\text{KL}_2(\mathbb{Q})}{n\lambda} + \frac{\lambda}{2} \bar{V}_n(\mathbb{Q}), \tag{32}$$

where the quantities are defined in Theorem 1 and the learning policies $\pi_\theta$ are defined as softmax policies as

$$\pi_\theta^{\text{SOF}}(a|x) = \frac{\exp(\phi(x)^\top \theta_a)}{\sum_{a' \in \mathcal{A}} \exp(\phi(x)^\top \theta_{a'})}, \tag{33}$$

To optimize (32), we employ the local reparameterization trick [Kingma et al., 2015]. Precisely, we set $\mathbb{Q} = \mathcal{N}\left(\mu, \sigma^2 I_{dK}\right)$ where $\mu \in \mathbb{R}^{dK}$ and $\sigma > 0$ are learnable parameters. Then roughly speaking, the terms $\mathbb{E}_{\theta \sim \mathbb{Q}}\left[\hat{R}(\pi_\theta, S)\right]$, $B_n(\mathbb{Q})$ and $\bar{V}_n(\mathbb{Q})$ in (32) are of the form $\mathbb{E}_{\theta \sim \mathcal{N}(\mu, \sigma^2 I_{dK})}\left[f(\pi_\theta^{\text{SOF}}(a|x))\right]$ for some function $f$ and they can be rewritten as

$$\mathbb{E}_{\theta \sim \mathcal{N}(\mu, \sigma^2 I_{dK})}\left[f(\pi_\theta^{\text{SOF}}(a|x))\right] = \mathbb{E}_{\theta \sim \mathcal{N}(\mu, \sigma^2 I_{dK})}\left[f\left(\frac{\exp(\phi(x)^\top \theta_a)}{\sum_{a' \in \mathcal{A}} \exp(\phi(x)^\top \theta_{a'})}\right)\right],$$

$$= \mathbb{E}_{\epsilon \sim \mathcal{N}(0, I_K)}\left[f\left(\frac{\exp(\phi(x)^\top \mu_a + \sigma \epsilon_a)}{\sum_{a' \in \mathcal{A}} \exp(\phi(x)^\top \mu_{a'} + \sigma \epsilon_{a'})}\right)\right],$$

where we use in the second equality the fact that $\|\phi(x)\|_2 = 1$ in our experiments since we normalized features. Then the above expectation can be approximated as

$$\mathbb{E}_{\theta \sim \mathcal{N}(\mu, \sigma^2 I_{dK})}\left[f(\pi_\theta^{\text{SOF}}(a|x))\right] \approx \frac{1}{S} \sum_{i \in [S]} f\left(\frac{\exp(\phi(x)^\top \mu_a + \sigma \epsilon_{i,a})}{\sum_{a' \in \mathcal{A}} \exp(\phi(x)^\top \mu_{a'} + \sigma \epsilon_{i,a'})}\right), \qquad \epsilon_i \sim \mathcal{N}(0, I_K), \forall i \in [S].$$

for some $S \geq 1$. Similarly, the gradients are approximated as

$$\nabla_{\mu, \sigma} \mathbb{E}_{\theta \sim \mathcal{N}(\mu, \sigma^2 I_{dK})}\left[f(\pi_\theta^{\text{SOF}}(a|x))\right] \approx \frac{1}{S} \sum_{i \in [S]} \nabla_{\mu, \sigma} f\left(\frac{\exp(\phi(x)^\top \mu_a + \sigma \epsilon_{i,a})}{\sum_{a' \in \mathcal{A}} \exp(\phi(x)^\top \mu_{a'} + \sigma \epsilon_{i,a'})}\right), \quad \epsilon_i \sim \mathcal{N}(0, I_K), \forall i \in [S].$$

## C.3 BOUND OPTIMIZATION WHEN IW REGULIZATION IS LINEAR

In this section, the learned policy $\hat{\pi}_n$ is obtained by optimizing the following objective derive from the bound in Corollary 2

$$\operatorname*{argmax}_{\mathbb{Q}} \hat{R}(\pi_{\mathbb{Q}}, S) + \sqrt{\frac{\text{KL}_1(\mathbb{Q})}{2n} + B_n(\pi_{\mathbb{Q}})} + \frac{\text{KL}_2(\mathbb{Q})}{n\lambda} + \frac{\lambda}{2} \bar{V}_n(\pi_{\mathbb{Q}}), \tag{34}$$

where $\text{KL}_1(\mathbb{Q}) = D_{\text{KL}}(\mathbb{Q}\|\mathbb{P}) + \log\frac{4\sqrt{n}}{\delta}$, $\text{KL}_2(\mathbb{Q}) = D_{\text{KL}}(\mathbb{Q}\|\mathbb{P}) + \log(4/\delta)$, and

$$\bar{V}_n(\pi_{\mathbb{Q}}) = \frac{1}{n}\sum_{i=1}^n \mathbb{E}_{a\sim\pi_0(\cdot|x_i)}\left[\frac{\pi_{\mathbb{Q}}(a|x_i)}{h(\pi_0(a|x_i))^2}\right] + \frac{\pi_{\mathbb{Q}}(a_i|x_i)}{h(\pi_0(a_i|x_i))^2}c_i^2$$

$$B_n(\pi_{\mathbb{Q}}) = 1 - \frac{1}{n}\sum_{i=1}^n\sum_{a\in\mathcal{A}}\pi_0(a|x_i)\frac{\pi_{\mathbb{Q}}(a|x_i)}{h(\pi_0(a|x_i))}.$$

We optimize this objective over learning policies $\pi_{\mathbb{Q}}$ that are defined as Gaussian policies of the form

$$\pi_{\mu,\sigma}^{\text{GAUS}}(a|x) = \mathbb{E}_{\theta\sim\mathcal{N}(\mu,\sigma^2 I_d)}\left[\pi_\theta(a|x)\right], \qquad \text{where } \pi_\theta(a|x) = \mathbb{1}_{\{\text{argmax}_{a'\in\mathcal{A}}\phi(x)^\top\theta_{a'}=a\}}. \tag{35}$$

Note that these Gaussian policies satisfy the form $\pi_{\mathbb{Q}}(a|x) = \mathbb{E}_{\theta\sim\mathbb{Q}}[\pi_\theta(a|x)]$ where $\pi_\theta$ is binary, which required by Corollary 2. There is no expectation in the above objective and hence the method described in Appendix C.2 is no longer needed. All we need is to be able to compute the propensities $\pi_{\mathbb{Q}}(a|x)$ and gradients of the above objective with respect to $\mathbb{Q}$, which boils down to computing the gradient of $\pi_{\mathbb{Q}}$ with respect to $\mathbb{Q}$ since the objective above is linear in $\pi_{\mathbb{Q}}$. Computing propensities and gradients is done as follows. First, Sakhi et al. [2023] showed that (35) can be written as

$$\pi_{\mu,\sigma}^{\text{GAUS}}(a|x) = \mathbb{E}_{\epsilon\sim\mathcal{N}(0,1)}\left[\prod_{a'\neq a}\Phi\left(\epsilon + \frac{\phi(x)^\top(\mu_a-\mu_{a'})}{\sigma\|\phi(x)\|}\right)\right],$$

where $\Phi$ is the cumulative distribution function of a standard normal variable. Then, the propensities are approximated as

$$\pi_{\mu,\sigma}^{\text{GAUS}}(a|x) \approx \frac{1}{S}\sum_{i\in[S]}\prod_{a'\neq a}\Phi\left(\epsilon_i + \frac{\phi(x)^\top(\mu_a-\mu_{a'})}{\sigma\|\phi(x)\|}\right), \qquad \epsilon_i\sim\mathcal{N}(0,1), \forall i\in[S]. \tag{36}$$

Similarly, the gradient reads

$$\nabla_{\mu,\sigma}\pi_{\mu,\sigma}^{\text{GAUS}}(a|x) = \mathbb{E}_{\epsilon\sim\mathcal{N}(0,1)}\left[\nabla_{\mu,\sigma}\prod_{a'\neq a}\Phi\left(\epsilon + \frac{\phi(x)^\top(\mu_a-\mu_{a'})}{\sigma\|\phi(x)\|}\right)\right],$$

which can be approximated as

$$\nabla_{\mu,\sigma}\pi_{\mu,\sigma}^{\text{GAUS}}(a|x) = \frac{1}{S}\sum_{i\in[S]}\nabla_{\mu,\sigma}\prod_{a'\neq a}\Phi\left(\epsilon_i + \frac{\phi(x)^\top(\mu_a-\mu_{a'})}{\sigma\|\phi(x)\|}\right),$$

The results are presented in Figure 4 and are generally consistent with the conclusions drawn in Section 5. The main distinction is that when utilizing linear IW regularizations and the approach detailed in Corollary 2, all methods exhibit better performance compared to when they are optimized using the theoretical bound in Theorem 1, which is applicable to potentially non-linear IW regularizations. This improvement is attributed to the reduction of variance achieved by removing the expectation $\mathbb{E}_{\theta\sim\mathbb{Q}}[\cdot]$ from the bound and employing Gaussian policies.

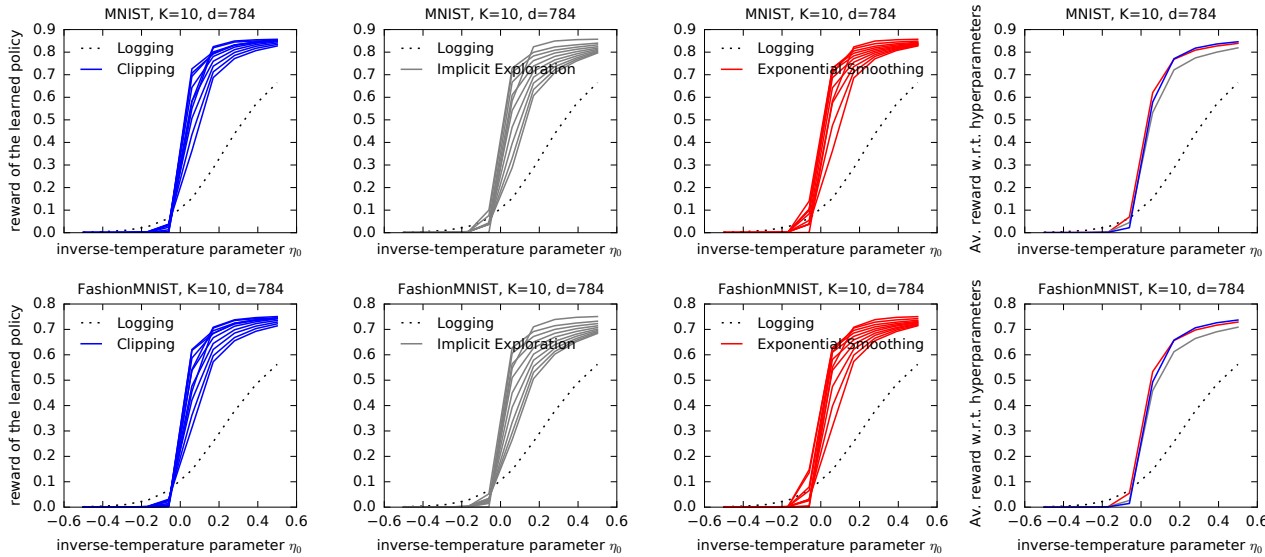

Figure 4: Performance of the policy learned by optimizing the bound in Corollary 2 for different IW regularizations. The $x$-axis reflects the quality of the logging policy $\eta_0 \in [-0.5, 0.5]$. In the first three columns, we plot the reward of the learned policy using a fixed IW regularization technique (`Clip`, `IX`, or `ES` as defined in (4)) for various values of its hyperparameter within $[0, 1]$. In the last column, we report the mean reward across these hyperparameter values.

## C.4 COMPARING HEURISTICS

We also compared our **Heuristic Optimization** (15) with the $L_2$ heuristic from London and Sandler [2019] and found that both heuristics exhibit identical performance (red and blue colors overlap in this plot).

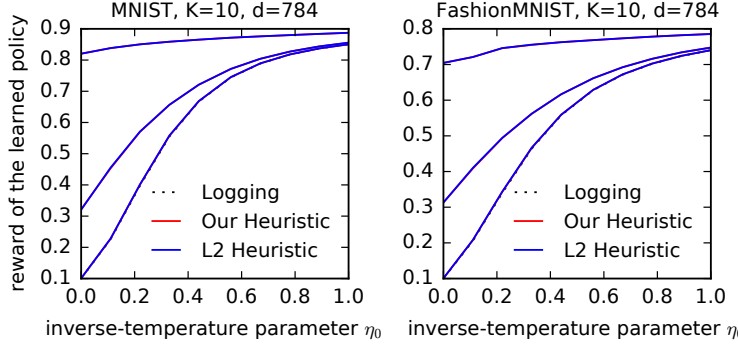

Figure 5: Performance of the learned policy with two learning principles (our **Heuristic Optimization** (15) and the $L_2$ heuristic in London and Sandler [2019] with varying values of their hyperparameters in a grid within $[10^{-5}, 10^{-3}]$) using the `Clip` IPS risk estimator in (4) with fixed $\tau = 1/\sqrt[4]{n}$.

## C.5 TIGHTNESS OF THE BOUND

We assess the tightness of our bound for a fixed IW regularization on the `MNIST` dataset. Specifically, we consider the `Clip` method as defined in (4), which regularizes the IW as $\hat{w}(x, a) = \frac{\pi(a|x)}{\max(\pi_0(a|x), \tau)}$. We apply Corollary 2 to this estimator by setting $h(p) = \max(p, \tau)$ and evaluate the bound at the learned policy for different values of $\tau$. The results are plotted in Figure 6. Generally, the bound is loose when the logging policy performs poorly, i.e., when $\eta_0 < 0.2$, and it tightens as the performance of the logging policy improves, i.e., as $\eta_0$ increases. The value of $\tau$ affects the bound tightness, but the impact is not very significant in the sense that there is no choice of $\tau$ that leads to a consistently loose bound, irrespective of the logging policy.

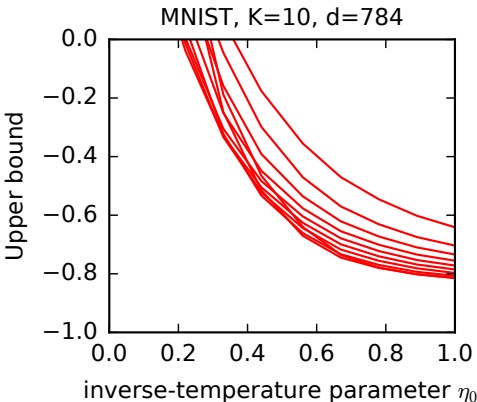

Figure 6: Tightness of the bound in Corollary 2 applied to `Clip`-IPS in (4) with varying values of hyperparameter $\tau$.